# Offline Reinforcement Learning with Penalized Action Noise Injection

## Abstract

Offline reinforcement learning (RL) seeks to optimize policies from fixed datasets, enabling deployment in domains where environment interaction is costly or unsafe. A central challenge in this setting is the overestimation of out-of-distribution (OOD) actions, which arises when Q-networks assign high values to actions absent from the dataset. To address this, we propose Penalized Action Noise Injection (PANI), a lightweight Q-learning approach that perturbs dataset actions with controlled noise to increase action-space coverage while introducing a penalty proportional to the noise magnitude to mitigate overestimation. We theoretically show that PANI is equivalent to Q-learning on a Noisy Action Markov Decision Process (NAMDP), providing a principled foundation for its design. Importantly, PANI is algorithm-agnostic and requires only minor modifications to existing off-policy and offline RL methods, making it broadly applicable in practice. Despite its simplicity, PANI achieves substantial performance improvements across various offline RL benchmarks, demonstrating both effectiveness and practicality as a drop-in enhancement.

## 1 Introduction

Reinforcement learning (RL) enables agents to learn decision making policies through interaction with an environment. While effective in many domains, real world applications such as healthcare and autonomous driving often restrict such interaction due to safety and cost concerns. Offline RL addresses this limitation by training policies on precollected datasets, removing the need for online exploration. Despite this advantage, offline RL remains challenged by the overestimation of out of distribution (OOD) actions, which can degrade policy performance at deployment.

To mitigate the OOD problem, recent approaches have explored the generative capabilities of diffusion models. Methods such as Diffusion-QL (Wang et al., 2022) and SfBC (Chen et al., 2022) use diffusion models to construct behavior cloning policies that support $Q$-learning. Other techniques, including QGPO (Lu et al., 2023), incorporate $Q$-value feedback to guide action sampling, while DTQL (Chen et al., 2024) and SRPO (Chen et al., 2023a) apply diffusion-based regularization instead of relying on generative policies. Although these methods achieve strong empirical results.

Motivated by the success of diffusion-based methods, which leverage multi-scale noise perturbations to provide reliable learning signals even in low-density regions of the data distribution, we propose Penalized Action Noise Injection (PANI). PANI is a lightweight method that perturbs actions from offline datasets with controlled noise and penalizes them according to the noise magnitude. In doing so, PANI broadens the coverage of $Q$-network updates across the action space, thereby mitigating overestimation errors in $Q$-learning while maintaining computational efficiency. While most offline RL algorithms inherently rely on the neural network's generalization to evaluate unseen actions, which often leads to overestimation in regions with limited or no data coverage, PANI enforces updates across the entire action space through noise-injected perturbations, which makes it distinctive in this regard among offline RL methods.

PANI is broadly compatible with existing off-policy and offline RL algorithms, requiring only minor modifications to integrate with methods such as IQL (Kostrikov et al., 2021), TD3 (Fujimoto et al., 2018), and even generative model-based approaches like QGPO (Lu et al., 2023). Our contributions are threefold. First, we introduce **Penalized Action Noise Injection (PANI)**, a simple yet theoretically

grounded method that enables $Q$-network updates over a broader region of the action space using only offline data. Second, we formalize the **Noisy Action MDP (NAMDP)**, a modified Markov Decision Process induced by noise injection, and provide a theoretical analysis of the resulting $Q$-values to show that it mitigates out-of-distribution (OOD) value overestimation. Third, we propose a **Hybrid Noise Distribution**, derived from the NAMDP-based analysis, which further improves performance and stability across tasks.

## 2 PRELIMINARIES

RL provides a foundational framework for solving sequential decision-making problems, where an agent learns to optimize its actions through interactions with an environment. Formally, RL problems are modeled as Markov Decision Processes (MDPs), defined by $(\mathcal{S}, \mathcal{A}, R, P, \gamma)$, where $\mathcal{S}$ is the state space, $\mathcal{A}$ is the action space, $R$ is the reward function, $P$ is the transition probability distribution, and $\gamma \in (0, 1)$ is the discount factor. The primary objective in RL is to find a policy $\pi$, which maps states to a probability distribution over actions, maximizing the expected cumulative discounted reward: $\eta(\pi) = \mathbb{E}_\pi \left[ \sum_{t=0}^\infty \gamma^t r_t \right]$, where $r_t$ denotes the reward received at time $t$. This is typically achieved by iteratively refining the policy to approach the optimal policy $\pi^* = \arg\max_\pi \eta(\pi)$. One way to evaluate the policy's performance is to estimate the action value function $Q^\pi(s, a)$, which measures the expected cumulative reward starting from state $s$, taking action $a$, and subsequently following policy $\pi$: $Q^\pi(s, a) = \mathbb{E}_\pi \left[ \sum_{t=0}^\infty \gamma^t r_t \mid s_0 = s, a_0 = a \right]$.

**Offline RL**    Offline reinforcement learning (RL) aims to learn a policy from a fixed dataset without further environment interaction. A key challenge is the overestimation of unseen, out-of-distribution (OOD) actions, whose $Q$-values remain uncorrected due to the lack of data. In deep RL, the standard practice of alternating value maximization and bootstrapped updates can amplify this issue. While online RL mitigates overestimation through continuous data collection, offline RL lacks this feedback, causing policies to overprioritize OOD actions and perform poorly at deployment if left unaddressed.

A **detailed discussion of related work** is provided in the Appendix A.

## 3 Q-VALUE OVERESTIMATION PROBLEM IN OFFLINE RL

Out-of-distribution (OOD) error poses a major challenge not only in offline reinforcement learning (RL) but also in generative modeling. A representative example is score matching (Hyvärinen & Dayan, 2005), which estimates the score function $\nabla_x \log p(x)$ of a distribution $p$ based on sample data. However, in regions where data is sparse, score estimates become unreliable due to insufficient updates, leading to significant estimation errors (Song & Ermon, 2019).

To address this issue, Denoising Score Matching (DSM) (Vincent, 2011) was introduced. Instead of directly estimating the score of the original data distribution, DSM learns the score function of a noise-perturbed version of the data. By injecting noise into samples, it enables learning signals even in low-density regions, helping mitigate errors in score estimation. Specifically, the score network $s_\theta$ is trained to minimize the following objective:

$$J_{\text{DSM}}(\theta) = \mathbb{E}_{x \sim p, \tilde{x} \sim q_\sigma(\cdot|x)} \left[ \|s_\theta(\tilde{x}) - \nabla \log q_\sigma(\tilde{x} \mid x)\|_2^2 \right],$$

where $q_\sigma(\cdot \mid x)$ is a noise distribution, typically Gaussian: $\mathcal{N}(x, \sigma^2 \mathbf{I})$.

However, DSM still tends to produce unreliable estimates at lower noise levels, where updates rely on sparse data. At higher noise levels, the added noise can obscure important data structures, reducing the accuracy of score estimation and degrading downstream performance.

Diffusion models extend the principles of DSM by learning the score of the noisy distribution through a multi-step denoising process (Song et al., 2020). These models are trained to progressively denoise data corrupted by noise at varying levels, corresponding to different time steps $t$. Specifically, by minimizing the following objective:

$$J(\theta) = \mathbb{E}_{t \sim \mathcal{U}(0,1), \epsilon \sim \mathcal{N}(0,\mathbf{I})} \left[ \|\epsilon_\theta(z_t, t) - \epsilon\|_2^2 \right] \quad \text{where} \quad z_t = \alpha_t x + \sigma_t \epsilon, \text{ thus, } z_t \sim \mathcal{N}(\alpha_t x, \sigma_t^2 \mathbf{I}).$$

This formulation can also be viewed from the perspective of DSM. Defining $q_{\sigma_t}(\cdot|x) \sim \mathcal{N}(\alpha_t x, \sigma_t^2 \mathbf{I})$:

$$\|\epsilon_\theta(z_t, t) - \epsilon\|_2^2 = \sigma_t^2 \|s_\theta(z_t, t) - \nabla \log q_{\sigma_t}(z_t|x)\|_2^2$$

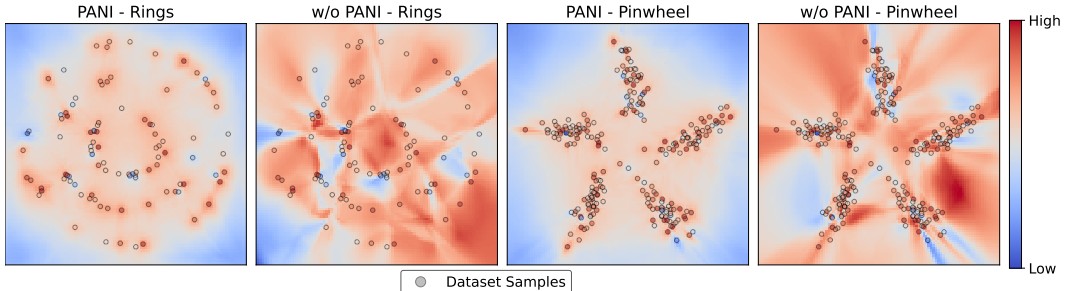

Figure 1: Visualization of learned $Q$-values on toy datasets. Each pair compares models trained with and without PANI on Rings (left) and Pinwheel (right). Background shows $Q$-values (red: high, blue: low); circles represent dataset actions, colored by their rewards.

we can rewrite the objective as:

$$J(\theta) = \mathbb{E}_{t \sim \mathcal{U}(0,1), z_t \sim q_{\sigma_t}(\cdot|x)} \left[ \sigma_t^2 \| s_\theta(z_t, t) - \nabla \log q_{\sigma_t}(z_t|x) \|_2^2 \right].$$

This training framework uses DSM to approximate the score function over multiple noise scales, combining the detailed signals at low noise levels with the robustness of higher noise. Moreover, training iteratively across noise levels can be seen as a form of data augmentation, as it exposes the model to a broader range of input variations during learning (Kingma & Gao, 2024).

In offline RL, analogous to score matching, limited coverage in the action space can hinder $Q$-network updates in low-density regions, potentially leading to overestimation of unseen or rarely sampled actions. In particular, actions absent from the dataset receive no direct learning signal, making them prone to out-of-distribution overestimation. To address this, we take inspiration from DSM and propose Penalized Noisy Action Injection, which injects noise into actions during training to encourage updates in underrepresented regions of the action space. Furthermore, motivated by the use of multi-scale noise in diffusion models, we introduce a hybrid noise distribution that combines various noise levels to enhance robustness. The effectiveness of our approach is illustrated in Figure 1.

## 4 PENALIZED ACTION NOISE INJECTION

In this section, we present Penalized Action Noise Injection (PANI), a simple yet effective method for extending value-based RL algorithms to better handle out-of-distribution actions. Given a dataset $\mathcal{D}$, conventional value-based methods typically optimize the following objective:

$$J(\theta) = \mathbb{E}_{(s,a,s') \sim \mathcal{D}} \left[ (Q_\theta(s, a) - y(s, a, s'))^2 \right],$$

where $y$ is the target value derived from the Bellman equation. In offline RL, updates are limited to actions in $\mathcal{D}$, leaving the $Q$-network prone to overestimating unseen actions.

To mitigate the overestimation of unseen actions, we inject noise into dataset actions, allowing the $Q$-function to be updated on perturbed actions while penalizing $Q$-values according to the squared distance from the original action. Specifically, we modify the standard update objective as follows:

$$\bar{J}(\theta) = \mathbb{E}_{(s,a,s') \sim \mathcal{D}, \bar{a} \sim q_\sigma(\cdot|a)} \left[ \left( Q_\theta(s, \bar{a}) - y(s, a, s') - \|a - \bar{a}\|_2^2 \right)^2 \right],$$

where the penalized target value is defined by $\bar{y}(s, a, s', \bar{a}) = y(s, a, s') - \|a - \bar{a}\|_2^2$. Here, the penalty term $\|a - \bar{a}\|_2^2$ discourages the network from assigning high $Q$-values to actions that deviate significantly from dataset actions.

Thus, our method can be implemented with minimal changes by sampling actions from the dataset, injecting noise to obtain perturbed actions $\bar{a}$, and updating the $Q$-network using the penalized target value. This procedure integrates seamlessly with various offline RL algorithms, requiring only minor modifications to the $Q$-update step (see Appendix C for details).

**Remark** It is worth noting that PANI operates entirely in the offline setting, unlike prior RL work where noise is widely used for representation learning (Laskin et al., 2020; Sinha et al., 2022) or simulation-based control (Qiao et al., 2021). PANI's distinct approach lies in employing penalized action noise to control $Q$-values of function approximators for unseen actions, rather than augmenting data or enhancing state representations.

# 5   NOISY ACTION MARKOV DECISION PROCESS

To understand how noise affects learning, we formalize the PANI objective as $Q$-learning within a new Markov Decision Process, the Noisy Action MDP (NAMDP), and analyze its properties to provide theoretical insights into the approach.

We begin by formally defining the noise distribution, which plays a crucial role in the formulation of the NAMDP. The noise distribution determines how noise is injected into the action space, influencing the updates in the $Q$-network and the resulting policy behavior.

**Definition 5.1** (Noise Distribution). A noise distribution $q_\sigma$ is a distribution parameterized by $a \in \mathcal{A}$ and a noise scale $\sigma > 0$, with support $\text{supp}(q_\sigma)$ such that the action space $\mathcal{A}$ is a subset of its support.

For example, a Gaussian noise distribution $q_\sigma(\bar{a} \mid a) = \mathcal{N}(\bar{a} \mid a, \sigma^2 \mathbf{I})$ satisfies these requirements.

Given a sample $\bar{a}$, the probability that is generated from $a$ under the distribution $p$, incorporating the noise distribution $q_\sigma$, can be expressed as:

$$p(\bar{a} \mid a, \sigma) = \frac{p(a) q_\sigma(\bar{a} \mid a)}{\int p(a) q_\sigma(\bar{a} \mid a)\, da}. \quad (1)$$

The denominator in this equation represents the convolution of the noise distribution $q_\sigma$ with the given distribution $p$:

$$p_\sigma(\bar{a}) = \int p(a) q_\sigma(\bar{a} \mid a)\, da.$$

If the noise distribution $q_\sigma$ is reasonable, the probability $p(\bar{a} \mid a, \sigma)$ tends to be higher for $\bar{a}$ near $a$ and lower for those farther away. This behavior is illustrated in Figure 2.

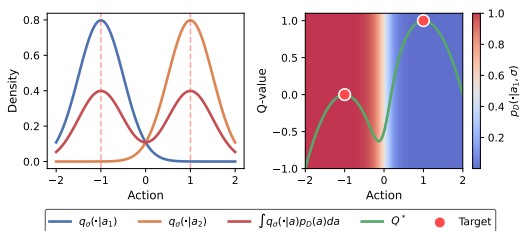

Figure 2: **Left:** Noise distributions and the resulting noised distribution. **Right:** $Q$-value predictions under the NAMDP, with the background color representing $p_D(\bar{a} \mid a_1, \sigma)$. Note that $a_1 = -1$, $a_2 = 1$, with rewards $r(a_1) = 0$, $r(a_2) = 1$. The green curve shows the groundtruth $Q$-values.

We define the Noisy Action MDP using the weight function $p_D(\bar{a} \mid s, a, \sigma)$, which, under a reasonable noise distribution, assigns higher weight to actions near $a$.

**Definition 5.2** (Noisy Action MDP). Given a noise distribution $q_\sigma$, a finite dataset $\mathcal{D} = \{(s_i, a_i, r_i, s_i')\}_i$ and a dataset distribution $p_D$, the NAMDP is defined as an MDP $(\mathcal{S}, \mathcal{A}, R_\sigma, P_\sigma, \gamma)$, where:

$$P_\sigma(s' \mid s, \bar{a}) = \int_{\mathcal{A}} p_D(s' \mid s, a)\, p_D(\bar{a} \mid s, a, \sigma)\, da,$$

$$R_\sigma(s, \bar{a}) = \int_{\mathcal{A}} p_D(\bar{a} \mid s, a, \sigma)\left(R(s, a) - \|a - \bar{a}\|_2^2\right) da.$$

Using this definition, we now show that minimizing the PANI objective is equivalent to learning the $Q$-value function in the NAMDP. This result provides a formal grounding for PANI within a modified decision process, enabling theoretical analysis based on the structure of the NAMDP.

**Theorem 5.3** (PANI Objective). *Suppose that the function $Q$ minimizes the following objective:*

$$\mathbb{E}_{a \sim p_D(\cdot \mid s),\, \bar{a} \sim q_\sigma(\cdot \mid a)}\left[(Q(s, \bar{a}) - \bar{y}(s, a, \bar{a}))^2\right],$$

*where the target value $\bar{y}(s, a, \bar{a})$ is defined as:*

$$\mathbb{E}_{s' \sim p_D(\cdot \mid s, a),\, \tilde{a} \sim \pi(\cdot \mid s')}\left[R(s, a) - \|a - \bar{a}\|_2^2 + \gamma Q^\pi(s', \tilde{a})\right].$$

*Then, the function $Q$ is the $Q$-value function of $\pi$ in the NAMDP.*

This offers a key insight into the behavior of PANI. As established in Definition 5.1, the noised training distribution spans the entire action space, mitigating the overestimation of $Q$-values typically caused by bootstrapped updates using out-of-distribution (OOD) actions in offline RL (Kumar et al., 2019). This directly leads to the result in the preceding theorem, demonstrating that minimizing the PANI objective yields the exact $Q$-value function of the NAMDP.

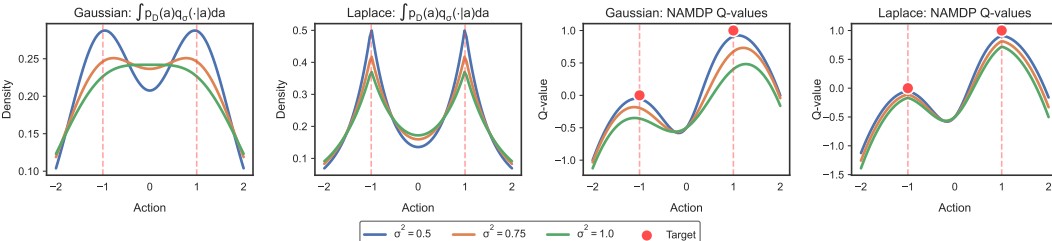

Figure 3: Comparison of noise distributions (Gaussian and Laplace) and their impact on NAMDP. The left two plots show the noisy action distributions $\int p_D(a)q_\sigma(\cdot \mid a)\, da$ for Gaussian (leftmost) and Laplace (second from left) distributions across different noisy levels ($\sigma^2 = 0.5, 0.75, 1.0$). The right two plots illustrate the corresponding NAMDP ground-truth $Q$-values under these distributions.

An interesting trade-off emerges as the noise level $\sigma$ increases. On one hand, the balanced action coverage induced by $q_\sigma$ accelerates the convergence of $Q$; on the other hand, higher noise levels cause the NAMDP to deviate further from the original MDP, introducing additional bias. Further theoretical results, including error bounds and an analysis showing how PANI discourages the selection of OOD actions, are provided in Appendix B.

## 6    NOISE DISTRIBUTION SELECTION

The choice of noise distribution plays a key role in the performance of PANI. As $\sigma \to 0$, the effect of PANI diminishes, leaving the overestimation problem unaddressed. When the injected noise is too small, perturbed actions remain close to the original dataset actions, and the number of samples required to sufficiently cover the action space increases sharply. As a result, the $Q$-network receives limited updates in low-density regions, allowing overestimated $Q$-values for unseen actions to persist. In contrast, we next examine the side effects associated with high noise levels.

Consider the formulation of $p_D(\bar{a} \mid s, a, \sigma)$ in Eq. (1), where the denominator is given by $p_\sigma(\bar{a} \mid s) = \int p_D(a \mid s)q_\sigma(\bar{a} \mid a)da$. In practice, since the dataset is finite, this results in $p_\sigma(\bar{a} \mid s)$ being a finite mixture of noise distributions centered at observed actions. As the noise level $\sigma$ increases, the modes of this mixture tend to shift away from the original action modes. This shift can cause the distribution to place non-negligible probability mass on actions far from the dataset support, inadvertently distorting the weight function $p_D(\bar{a} \mid s, a, \sigma)$ to emphasize unreliable actions, which amplifies OOD overestimation.

For example, when using a Gaussian noise distribution, the leftmost plot in Figure 3 shows how increasing the noise level $\sigma$ (the standard deviation) affects the noisy distribution $p_\sigma$. At low noise levels ($\sigma^2 = 0.5$), the distribution retains distinct modes that reflect the dataset's original action structure. As $\sigma$ increases ($\sigma^2 \geq 0.75$), these modes blur and merge, obscuring the boundaries between actions. The right plot shows how this mode collapse distorts the ground-truth $Q$-function in the NAMDP: with high noise levels, $Q$-values are overestimated in regions far from the dataset.

High noise levels can also lead to sample inefficiency, resulting in ineffective $Q$-network updates. In typical RL settings with bounded action spaces, using an unbounded noise distribution such as Gaussian can cause sampled actions to fall outside the valid range, producing invalid or uninformative targets. Additionally, high noise levels reduce the precision of action sampling. For example, when two actions $a_1$ and $a_2$ are far apart, the likelihood of sampling a noisy action $\bar{a}$ near either becomes nearly uniform. This flattens the sampling distribution, increases variance in the estimated targets, and ultimately reduces learning stability. These observations highlight the importance of carefully selecting and tuning the noise distribution in PANI to balance expressiveness and stability.

### 6.1    DISTRIBUTION SELECTION

Above, we analyzed the trade-offs associated with high and low noise levels. Here, we explore strategies for selecting noise distributions that balance these trade-offs. By tailoring the noise distribution, we aim to maintain broad coverage of the action space while avoiding issues such as mode collapse and sample inefficiency.

**Leptokurtic Distributions**  Leptokurtic distributions, which have sharper peaks and heavier tails than Gaussians, offer favorable properties for noise injection. The Laplace distribution is a representative example. Its sharp peak helps preserve distinct modes, mitigating the mode shift problem under high noise. Meanwhile, the heavier tails increase the likelihood of sampling distant actions, supporting updates in less-visited regions of the action space.

To further analyze the effect of noise shape, we compare Gaussian and Laplace distributions under equal variance. As shown in Figure 3, the comparison highlights how distribution shape affects mode preservation and the resulting $Q$-values in the NAMDP. The results show that even under fixed variance, the shape of the noise distribution has a significant impact on mode preservation and $Q$-value quality. These findings suggest that leptokurtic distributions are well-suited for PANI, as they mitigate mode collapse at high noise levels and improve coverage at low noise levels.

While leptokurtic distributions help alleviate high-noise issues such as mode collapse and instability they remain sensitive to the choice of noise scale. To address this limitation and improve robustness across a broader range of noise levels, we design the Hybrid Noise Distribution, which combines the concentrated mass of leptokurtic distributions with the broader coverage of a uniform component.

**Hybrid Noise Distribution**  The hybrid noise distribution is designed to maintain robust performance across varying noise scales by combining two complementary components: a uniform mixture for broad exploration and an exponential scaling mechanism to induce leptokurtic behavior. Its construction involves two key steps.

First, to address sample inefficiency at high noise levels, we define a mixture of the original noise and a uniform distribution over the action space:

$$q_t^{\mathcal{A}}(\bar{a} \mid a) = \alpha(t)\,\mathcal{U}(\bar{a} \mid \mathcal{A}) + (1 - \alpha(t))\,q_t(\bar{a} \mid a),$$

where $\mathcal{U}(\bar{a} \mid \mathcal{A})$ denotes the uniform distribution over $\mathcal{A}$, and $\alpha(t) = \min(t, 1)$. As $t \to 1$, the distribution transitions smoothly to uniform, increasing coverage and reducing the likelihood of out-of-bound or degenerate samples.

To further improve robustness, we build on the motivation presented in Section 3, where diffusion-based methods leverage multi-scale noise to stabilize learning. Specifically, exponential scaling of Gaussian noise is known to increase kurtosis (West, 1987), which helps preserve mode structure at high noise, reducing instability across noise levels.

Based on this insight, we define the hybrid noise distribution as:

$$q_\sigma^{\mathrm{hyb}}(\bar{a} \mid a) = \mathbb{E}_{\lambda \sim \mathcal{U}(\log \sigma, 0)}\left[q_{\exp(\lambda)}^{\mathcal{A}}(\bar{a} \mid a)\right],$$

where $t = \exp(\lambda)$. Here, $\sigma$ controls the overall noise level rather than serving as a standard deviation. This formulation combines the broad coverage of the uniform mixture with the scale-adaptive kurtosis induced by exponential sampling, leading to improved robustness.

In our experiments, we used a Gaussian base noise defined as $q_t(\bar{a} \mid a) = \mathcal{N}(\bar{a} \mid a, t^2\mathbf{I})$. This setup balances exploration with local structure, resulting in stable and effective $Q$-network updates. To evaluate its effectiveness, we compared the hybrid noise distribution with Gaussian and Laplace noise. As shown in Figure 4a, the hybrid distribution was more stable across noise levels, and we use it as the default in all subsequent experiments.

## 7 EXPERIMENTS

In this section, we present experimental results demonstrating the effectiveness of the Penalized Action Noise Injection (PANI) method. We first show that applying PANI to baseline algorithms such as TD3 (Fujimoto et al., 2018) and IQL (Kostrikov et al., 2021) leads to significant performance improvements across various datasets and environments in the various benchmark. Next, we conduct a series of ablation studies to answer the following research questions: **Q1**. Does the Hybrid Noise Distribution provide more robust performance across different noise scales compared to Gaussian and Laplace noise? **Q2**. Is PANI computationally more efficient than diffusion-based methods? **Q3**. Does PANI effectively reduce OOD $Q$-value overestimation? **Q4**. Does PANI consistently improve performance when applied to other state-of-the-art offline RL algorithms?

Table 1: Average normalized scores following (Fu et al., 2020), reported on Gym-MuJoCo and AntMaze tasks. Results are computed over 5 random seeds with 10 trajectories per seed for Gym-MuJoCo, and 100 trajectories per seed for AntMaze. The $\pm$ symbol indicates standard error across training seeds. Bold numbers indicate the highest score for each task, and underlined values denote the second highest. Red text shows the performance gain relative to the original baseline.

| Dataset | Environment | Diffusion-policy | | | Diffusion-based | | Diffusion-free | | Ours | |
| | | SfBC | D-QL | QGPO | SRPO | DTQL | IQL | TD3+BC | IQL-AN | TD3-AN |
|---|---|---|---|---|---|---|---|---|---|---|
| medium | halfcheetah | 45.9 | 51.1 | 54.1 | 60.4 | 57.9 | 50.0 | 54.7 | 55.4 ± 0.3 | **61.5** ± 0.3 |
| medium | hopper | 57.1 | 90.5 | 98.0 | 95.5 | **99.6** | 65.2 | 60.9 | 98.4 ± 1.2 | 98.2 ± 0.9 |
| medium | walker2d | 77.9 | 87.0 | 86.0 | 84.4 | **89.4** | 80.7 | 77.7 | 87.5 ± 3.7 | 88.5 ± 0.6 |
| medium-replay | halfcheetah | 37.1 | 47.8 | 47.6 | 51.4 | 50.9 | 42.1 | 45.0 | 49.5 ± 0.4 | **53.3** ± 0.3 |
| medium-replay | hopper | 86.2 | 100.7 | 96.9 | 101.2 | 100.0 | 89.6 | 55.1 | 100.8 ± 0.4 | **102.3** ± 0.2 |
| medium-replay | walker2d | 65.1 | **95.5** | 84.4 | 84.6 | 88.5 | 75.4 | 68.0 | 88.8 ± 3.6 | 87.8 ± 6.3 |
| medium-expert | halfcheetah | 92.6 | **96.8** | 93.5 | 92.2 | 92.7 | 92.7 | 89.1 | 89.9 ± 2.3 | 96.4 ± 0.8 |
| medium-expert | hopper | 108.6 | **111.1** | 108.0 | 100.1 | 109.3 | 85.5 | 87.8 | 105.3 ± 3.7 | 108.8 ± 0.9 |
| medium-expert | walker2d | 109.8 | 110.1 | 110.7 | 114.0 | 110.0 | 112.1 | 110.4 | 109.6 ± 0.6 | **114.9** ± 0.2 |
| **Average** | | 75.6 | 88.0 | 86.6 | 87.1 | 88.7 | 77.0 | 72.1 | 87.2 (+10.2) | **90.2** (+18.1) |
| - | umaze | 92.0 | 93.4 | 96.4 | 97.1 | 92.6 | 83.3 | 66.3 | 91.2 ± 1.1 | **98.4** ± 0.5 |
| diverse | umaze | **85.3** | 66.2 | 74.4 | 82.1 | 74.4 | 70.6 | 53.8 | 68.0 ± 3.2 | 74.6 ± 4.9 |
| play | medium | 81.3 | 76.6 | 83.6 | 80.7 | 76.0 | 64.6 | 26.5 | 74.4 ± 3.7 | **83.8** ± 2.7 |
| diverse | medium | 82.0 | 78.6 | 83.8 | 75.0 | 80.6 | 61.7 | 25.9 | 75.2 ± 1.6 | **85.8** ± 1.2 |
| play | large | 59.3 | 46.4 | **66.6** | 53.6 | 59.2 | 42.5 | 0.0 | 49.4 ± 3.0 | 65.4 ± 2.7 |
| diverse | large | 45.5 | 56.6 | **64.8** | 53.6 | 62.0 | 27.6 | 0.0 | 52.8 ± 2.6 | 58.0 ± 5.1 |
| **Average (AntMaze)** | | 74.2 | 69.6 | **78.3** | 73.7 | 74.1 | 58.4 | 28.8 | 68.5 (+10.1) | 77.7 (+48.9) |

Table 2: Performance comparison on the Adroit pen tasks. Results are averaged over 5 random seeds with 100 trajectories per seed, and the $\pm$ symbol indicates the standard error across seeds. TD3-AN results are reported using this protocol, and baseline results are taken from the original papers.

| Dataset | Env | TD3-AN | CPQL | CPIQL | FQL | DTQL | DQL | CAC |
|---|---|---|---|---|---|---|---|---|
| cloned | pen | **93.6** ± 5.1 | 65.3 ± 1.1 | 57.4 ± 2.2 | 74.0 ± 3.9 | 81.3 ± 3.0 | 57.3 ± 11.9 | 50.1 ± 1.0 |
| human | pen | **80.6** ± 5.4 | 56.7 ± 2.2 | 48.0 ± 3.8 | 53.0 ± 2.1 | 64.1 ± 3.0 | 72.8 ± 9.6 | 63.4 ± 3.4 |

## 7.1 EVALUATION

We evaluate the effectiveness of PANI across a diverse set of offline RL benchmarks, including Gym-MuJoCo, AntMaze, Adroit, and OGBench. On Gym-MuJoCo and AntMaze (Table 1), we applied PANI to both the off-policy algorithm TD3 (Fujimoto et al., 2018) and the offline RL algorithm IQL (Kostrikov et al., 2021), yielding TD3-AN and IQL-AN. Both variants consistently improved over their original baselines and performed competitively with state-of-the-art diffusion-based methods such as DQL (Wang et al., 2022), SfBC (Chen et al., 2022), QGPO (Lu et al., 2023), DTQL (Chen et al., 2024), and SRPO (Chen et al., 2023a).

On the Adroit dataset (Table 2), we compared TD3-AN with several generative model-based approaches, including the consistency model methods CPQL and CPIQL (Chen et al., 2023b), the flow-based method FQL (Park et al., 2025), and the diffusion-based algorithms DTQL (Chen et al., 2024) and DQL (Wang et al., 2022). The PANI-enhanced variant TD3-AN achieved performance that was competitive with or superior to these advanced methods.

Finally, on the challenging OGBench benchmark (Table 3), specifically the antmaze-giant-navigate-singletask suite, we compared TD3-AN with IDQL (Hansen-Estruch et al., 2023), SRPO (Chen et al., 2023a), CAC (Ding & Jin, 2023), FAWAC, FBRAC, IFQL, and FQL (Park et al., 2025). TD3-AN achieved non-zero success rates in settings where most of these generative approaches failed, underscoring its effectiveness in this challenging regime. Taken together, these results demonstrate that PANI achieves strong generalization across diverse benchmarks, extending its applicability to a wide range of datasets and environments.

Table 3: Performance comparison on the antmaze-giant-navigate-singletask suite across five tasks. TD3-AN results are averaged over five random seeds, with standard errors reported. Performance metrics for all other baselines are taken from Park et al. (2025).

| Task | TD3-AN | IDQL | SRPO | CAC | FAWAC | FBRAC | IFQL | FQL |
|------|--------|------|------|-----|-------|-------|------|-----|
| Task 1 | **14.2** ± 6.5 | 0 ± 0.0 | 0 ± 0.0 | 0 ± 0.0 | 0 ± 0.0 | 0 ± 0.4 | 0 ± 0.0 | 4 ± 1.8 |
| Task 2 | 5.8 ± 4.4 | 0 ± 0.0 | 0 ± 0.0 | 0 ± 0.0 | 0 ± 0.0 | 4 ± 2.5 | 0 ± 0.0 | **9** ± 2.5 |
| Task 3 | **1.6** ± 0.7 | 0 ± 0.0 | 0 ± 0.0 | 0 ± 0.0 | 0 ± 0.0 | 0 ± 0.0 | 0 ± 0.0 | 0 ± 0.4 |
| Task 4 | **21.6** ± 10.4 | 0 ± 0.0 | 0 ± 0.0 | 0 ± 0.0 | 0 ± 0.0 | 9 ± 1.4 | 0 ± 0.0 | 14 ± 8.1 |
| Task 5 | 6.4 ± 2.2 | 0 ± 0.0 | 0 ± 0.0 | 0 ± 0.0 | 0 ± 0.0 | 6 ± 3.5 | 13 ± 3.2 | **16** ± 9.9 |

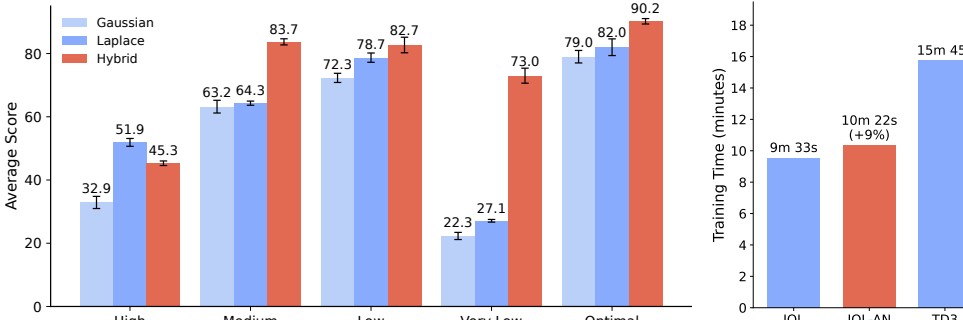

(a) Average score in Gym-MuJoCo, comparing different noise distributions.

(b) Training wall-clock time ($10^6$ steps).

Figure 4: Comparison of noise distribution and noise scale effects in Gym-MuJoCo. (a) Average score of TD3-AN comparing different noise distributions. High, Medium, Low, and Very Low represent progressively lower noise levels, with specific ranges adjusted for each distribution. Error bars indicate the standard error of the average score across 5 training seeds. Optimal represents the best-performing noise scale selected individually for each distribution. (b) Wall-clock training time for different algorithms, showing that the use of PANI incurs only a small overhead.

## 7.2 ABLATION STUDY

We now provide detailed analyses addressing the five research questions introduced above. Specifically, we examine the impact of different noise distributions, noise scales, and dataset action coverage on performance. We also evaluate the computational efficiency of PANI and assess its generalizability by applying it to other state-of-the-art offline RL algorithms.

**Noise Distribution** The choice of noise distribution is a critical factor in PANI. As discussed in Section 6, the hybrid noise distribution is specifically designed to address the sensitivity of performance to variations in noise scale. To evaluate its effectiveness, we compared it against both the Gaussian and Laplace Noise Distributions under identical hyperparameter settings, using different ranges of noise levels tailored to the characteristics of each distribution.

For the hybrid noise distribution, we tested a range of noise levels appropriate to its characteristics. Similarly, for the Gaussian and Laplace noise distributions, we selected noise levels that reflect their typical behavior and scale, ensuring a fair comparison across all methods. As shown in Figure 4a, the hybrid noise distribution outperformed the Gaussian and Laplace noise distributions. These results highlight the hybrid distribution's ability to balance robustness and coverage, making it well-suited for a variety of environments. Detailed settings for each noise scale, along with full results and learning curves, are provided in Appendix C.5.

**Compute Resource** We compared the training time of PANI-enhanced methods with their original counterparts. As shown in Figure 4b, PANI introduces minimal computational overhead while maintaining efficient training times. Detailed results, including hardware specifications, are provided in Appendix C.

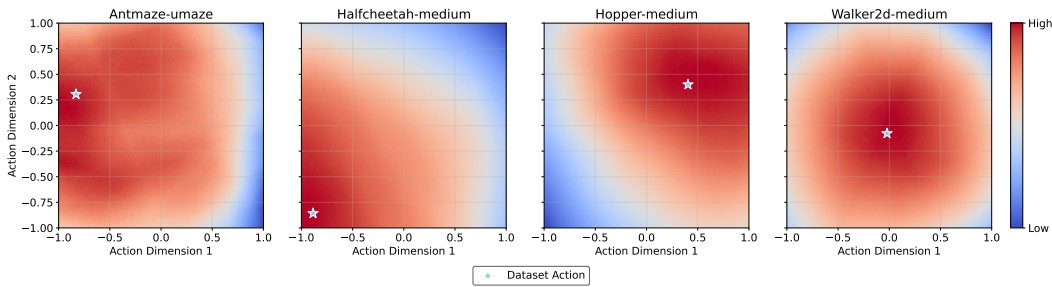

Figure 5: Learned $Q$-value landscapes in Gym-MuJoCo datasets. Colors represent $Q$-value magnitudes (red: high, blue: low), and the white star indicates the dataset action. These plots illustrate how the learned $Q$-functions evaluate dataset and surrounding actions across different environments.

| Dataset | TD3+BC | TD3-AN | IQL | IQL-AN |
|---|---|---|---|---|
| halfcheetah-expert | 0.014 | **0.002** | 0.014 | **0.001** |
| halfcheetah-medium | 0.112 | **0.004** | 0.211 | **0.006** |
| hopper-expert | 0.483 | **0.034** | 0.425 | **0.036** |
| hopper-medium | 0.379 | **0.036** | 0.360 | **0.042** |
| walker2d-expert | 0.322 | **0.000** | 0.327 | **0.001** |
| walker2d-medium | 0.316 | **0.002** | 0.147 | **0.004** |

(a) Measured probability of OOD overestimation $P(Q(s, \bar{a}) > Q(s, a))$ across datasets, comparing baseline and PANI-enhanced methods.

(b) Performance difference between QGPO and QGPO-AN.

Figure 6: Empirical evaluation of OOD overestimation reduction and performance improvement. (a) shows the probability of OOD overestimation $P(Q(s, \bar{a}) > Q(s, a))$, where $(s, a) \sim D$ and $\bar{a}$ is uniformly sampled, evaluated across different datasets (all 95% confidence intervals < 0.001). (b) shows the performance difference of QGPO-AN relative to QGPO across multiple tasks.

**Overestimation Mitigation** We evaluate the effectiveness of PANI in reducing OOD action overestimation. First, as shown in Figure 5, PANI predicts lower $Q$-values for actions that deviate from the dataset actions in Gym-MuJoCo, indicating reduced overestimation in out-of-distribution regions. Second, we measure the probability $P(Q(s, \bar{a}) > Q(s, a))$, where $a$ is a dataset action and $\bar{a}$ is uniformly sampled. As summarized in Table 6a, TD3-AN and IQL-AN significantly lower this probability across all datasets, demonstrating improved robustness compared to their base versions.

**PANI with Other Algorithms** We also applied PANI to QGPO (Lu et al., 2023), a diffusion-based $Q$-learning algorithm. Using the same hyperparameters $(\beta_Q, K)$ as in the original QGPO for fair comparison, QGPO-AN showed performance improvements, as shown in Figure 6b, demonstrating the generality of PANI. See Appendix C.3 for implementation details.

## 8 CONCLUSION

We introduced Penalized Action Noise Injection (PANI), a lightweight method designed to address out-of-distribution overestimation in offline RL. By perturbing dataset actions with controlled noise and penalizing them according to noise magnitude, PANI broadens $Q$-network updates across the action space with minimal modification to existing algorithms. Unlike most offline RL methods, which rely on neural network generalization to evaluate unseen actions and are thus prone to overestimation in low-coverage regions, PANI enforces updates across the entire action space, making it distinctive in this regard. Evaluations on various benchmarks demonstrate that PANI generalizes effectively across different datasets, environments, and algorithmic families. These results suggest that PANI is not only a practical tool for enhancing existing offline RL algorithms but also a broadly applicable principle for addressing a fundamental limitation of learning from fixed datasets.

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

APPENDIX

# A  RELATED WORKS

**Offline RL**   Various algorithms have been proposed to address the issue of OOD action overestimation in offline reinforcement learning. These approaches can be broadly categorized into three groups: (1) methods that enforce conservative $Q$-value estimates, (2) methods that apply penalties during policy extraction, and (3) methods that leverage generative models, either as policies or as regularizers, to guide action selection away from OOD actions.

Notable examples in the first category include CQL (Kumar et al., 2020), which explicitly trains the Q-network to assign lower values to OOD actions, and ensemble-based approaches such as EDAC (An et al., 2021) and RORL (Yang et al., 2022), which mitigate overestimated $Q$-values by aggregating predictions from multiple Q-networks.

In the second category, policy-level regularization techniques mitigate OOD actions through explicit penalties. AWAC (Nair et al., 2020) introduces KL regularization, while TD3+BC (Fujimoto & Gu, 2021) and ReBRAC (Tarasov et al., 2024) apply L2 penalties to keep the learned policy close to the behavior policy. Anti-exploration approaches such as TD3-CVAE (Rezaeifar et al., 2022), SAC-RND (Nikulin et al., 2023), and SAC-DRND (Yang et al., 2024) explicitly discourage selection of uncertain or unfamiliar actions.

The third category leverages generative models either as behavior policies or as regularizers during policy learning to avoid OOD actions. Early approaches such as BCQ (Fujimoto et al., 2019) use a VAE to generate filtered actions, avoiding OOD actions during policy improvement. More recent diffusion-based methods, including Diffusion-QL (Wang et al., 2022), SfBC (Chen et al., 2022), and IDQL (Hansen-Estruch et al., 2023), construct denoising-based behavior policies. SRPO (Chen et al., 2023a) and DTQL (Chen et al., 2024) regularize policy learning by constraining action selection to stay close to the dataset, thereby avoiding OOD actions. In contrast, QGPO (Lu et al., 2023) combines Q-guidance with generative sampling to guide the policy toward high-value actions.

In contrast to these approaches, PANI directly addresses OOD overestimation by modifying the Q-learning objective to include penalized updates on noise-injected actions.

**Data Augmentation in RL**   While noise-based augmentations are used in RL to enhance sample efficiency or generalization, most existing approaches focus on representation learning rather than addressing OOD overestimation.

For example, RAD (Laskin et al., 2020) applies image-based augmentations to observations in online RL to improve generalization. S4RL (Sinha et al., 2022) adapts this idea to the offline setting, applying RAD-style augmentation to states in static datasets. These methods target representation learning rather than correcting distributional mismatch in action values.

Some works consider action noise in the context of differentiable simulation. For instance, Qiao et al. (2021) use noisy actions to approximate gradients and transitions in simulator-based environments, enabling more efficient learning. In contrast, our method injects noise in the offline setting without access to a simulator, using it not to create synthetic targets but to penalize $Q$-values for out-of-distribution actions.

**Various Noise Scales and Leptokurtic Noise Distributions**   Noise variation and scaling have been extensively studied in generative modeling and denoising frameworks. MDNS (Li et al., 2023) explores training with diverse noise levels in denoising score matching, while NCSN (Song & Ermon, 2019) conditions training on continuous noise scales. Diffusion-based methods such as Score Distillation Sampling (Poole et al., 2022) leverage outputs across multiple noise levels to stabilize generation.

While most prior work uses Gaussian noise, recent studies explore alternative noise shapes to improve robustness. Heavy-Tailed Denoising Score Matching (HTDSM) (Deasy et al., 2021) and t-EDM (Pandey et al., 2024) employ leptokurtic noise distributions to better handle outliers and data sparsity.

Our method draws on these insights, but applies them in a context: penalizing noisy actions in the offline RL setting to systematically reduce OOD value overestimation.

# B    NOISY ACTION MARKOV DECISION PROCESS

In this section, we revisit the definition of the Noisy Action Markov Decision Process (NAMDP) and the associated noise distribution. We provide formal proofs for the key theorems introduced in the main text and offer additional theoretical insights under specific assumptions.

**Definition B.1** (Noise Distribution). A noise distribution $q_\sigma$ is a distribution parameterized by $a \in \mathcal{A}$ and a noise scale $\sigma > 0$, with support $\text{supp}(q_\sigma)$ such that the action space $\mathcal{A}$ is a subset of its support.

**Definition B.2** (Noisy Action MDP (NAMDP)). Given a noise distribution $q_\sigma$, a finite dataset $\mathcal{D} = \{(s_i, a_i, r_i, s_i')\}_i$ and a dataset distribution $p_D$, the NAMDP is defined as an MDP $(\mathcal{S}, \mathcal{A}, R_\sigma, P_\sigma, \gamma)$, where:

$$P_\sigma(s' \mid s, \bar{a}) = \int_{\mathcal{A}} p_D(s' \mid s, a)\, p_D(\bar{a} \mid s, a, \sigma)\, da, \tag{2}$$

$$R_\sigma(s, \bar{a}) = \int_{\mathcal{A}} p_D(\bar{a} \mid s, a, \sigma)\left(R(s, a) - \|a - \bar{a}\|_2^2\right) da, \tag{3}$$

$$p_D(\bar{a} \mid s, a, \sigma) = \frac{q_\sigma(\bar{a} \mid a) p_D(a|s)}{\int q_\sigma(\bar{a} \mid a) p_D(a|s)\, da}. \tag{4}$$

## B.1    CONNECTION BETWEEN PANI AND NAMDP

In this subsection, we present a key result that connects the PANI objective with the Noisy Action Markov Decision Process (NAMDP). Specifically, we show that Q-learning with PANI corresponds to learning the Q-function of the NAMDP under the defined noise distribution.

**Theorem B.3** (PANI Objective). *Suppose that the function $Q$ minimizes the following objective:*

$$\mathbb{E}_{a \sim p_D(\cdot|s),\, \bar{a} \sim q_\sigma(\cdot|a)} \left[\|Q(s, \bar{a}) - \bar{y}(s, a, \bar{a})\|_2^2\right], \tag{5}$$

*where the target value $\bar{y}(s, a, \bar{a})$ is defined as:*

$$\mathbb{E}_{s' \sim p_D(\cdot|s,a),\, \bar{a} \sim \pi(\cdot|s')} \left[R(s, a) - \|a - \bar{a}\|_2^2 + \gamma Q^\pi(s', \bar{a})\right].$$

*Then, the function $Q$ is the Q-value function of $\pi$ in the NAMDP.*

*Proof.* Let us derive the optimal Q-function for the NAMDP by applying the Euler equation for functionals (Gelfand et al., 2000). Consider a functional of the form:

$$J[u] = \int \cdots \int_{\mathbb{R}} F(x_1, \ldots, x_n, u) dx_1 \cdots dx_n \tag{6}$$

which depends on $n$ independent variables $x_1, \ldots, x_n$ and an unknown function $u$ of these variables. For the functional to be optimal, the following condition must hold:

$$F_u(x) = 0 \quad \text{for all} \quad x \tag{7}$$

The NAMDP objective eq. (5) can similarly be expressed as a functional:

$$J[Q] = \mathbb{E}_{\substack{a \sim p_D(\cdot|s) \\ \bar{a} \sim q_\sigma(\cdot|a)}} \left[ \left\| Q(s, \bar{a}) - \mathbb{E}_{\substack{s' \sim p_D(\cdot|s,a) \\ \bar{a} \sim \pi(\cdot|s')}} \left[R(s, a) - \|\bar{a} - a\|_2^2 + \gamma Q^\pi(s', \bar{a})\right] \right\|_2^2 \right] \tag{8}$$

$$= \int_{\mathbb{R}^n} \int_{\mathcal{A}} p_D(a|s) q_\sigma(\bar{a}|a)$$

$$\left\| Q(s, \bar{a}) - \mathbb{E}_{\substack{s' \sim p_D(\cdot|s,a) \\ \bar{a} \sim \pi(\cdot|s')}} \left[R(s, a) - \|\bar{a} - a\|_2^2 + \gamma Q^\pi(s', \bar{a})\right] \right\|_2^2 da\, d\bar{a} \tag{9}$$

$$= \int_{\mathbb{R}^n} F(\bar{a}_1, \ldots, \bar{a}_n, Q) d\bar{a} \tag{10}$$

where

$$F(\bar{a}_1, \ldots, \bar{a}_n, Q) = \tag{11}$$

$$\int_{\mathcal{A}} p_D(a|s) q_\sigma(\bar{a}|a) \left\| Q(s, \bar{a}) - \mathbb{E}_{\substack{s' \sim p_D(\cdot|s,a) \\ \bar{a} \sim \pi(\cdot|s')}} \left[R(s, a) - \|\bar{a} - a\|_2^2 + \gamma Q^\pi(s', \bar{a})\right] \right\|_2^2 da \tag{12}$$

Suppose that $Q^*$ is the minimizer of the NAMDP objective equation 5. By the Euler equation for the functional equation 6, we have:

$$F_{Q^*}(\bar{a}) = \tag{13}$$

$$2\int_{\mathcal{A}} p_D(a|s)q_\sigma(\bar{a}|a)\left(Q^*(s,\bar{a}) - \mathbb{E}_{\substack{s'\sim p_D(\cdot|s,a)\\ \bar{a}\sim\pi(\cdot|s')}}\left[R(s,a) - \|\bar{a}-a\|_2^2 + \gamma Q^\pi(s',\bar{a})\right]\right)da = 0 \tag{14}$$

Thus,

$$\int_{\mathcal{A}} p_D(a|s)q_\sigma(\bar{a}|a)Q^*(s,\bar{a})da \tag{15}$$

$$= \int_{\mathcal{A}} p_D(a|s)q_\sigma(\bar{a}|a)\mathbb{E}_{\substack{s'\sim p_D(\cdot|s,a)\\ \bar{a}\sim\pi(\cdot|s')}}\left[R(s,a) - \|\bar{a}-a\|_2^2 + \gamma Q^\pi(s',\bar{a})\right]da \tag{16}$$

$$\iff Q^*(s,\bar{a}) = \frac{\int_{\mathcal{A}} p_D(a|s)q_\sigma(\bar{a}|a)\mathbb{E}_{\substack{s'\sim p_D(\cdot|s,a)\\ \bar{a}\sim\pi(\cdot|s')}}\left[R(s,a) - \|\bar{a}-a\|_2^2 + \gamma Q^\pi(s',\bar{a})\right]da}{\int_{\mathcal{A}} p_D(a|s)q_\sigma(\bar{a}|a)da} \tag{17}$$

$$= \frac{\int_{\mathcal{S}}\int_{\mathcal{A}} p_D(a|s)q_\sigma(\bar{a}|a)p_D(s'|s,a)\mathbb{E}_{\bar{a}\sim\pi(\cdot|s')}\left[R(s,a) - \|\bar{a}-a\|_2^2 + \gamma Q^\pi(s',\bar{a})\right]dads'}{\int_{\mathcal{A}} p_D(a|s)q_\sigma(\bar{a}|a)da} \tag{18}$$

$$= \int_{\mathcal{S}}\int_{\mathcal{A}} p_D(s'|s,a)p_D(\bar{a}|s,a,\sigma)\mathbb{E}_{\bar{a}\sim\pi(\cdot|s')}\left[R(s,a) - \|\bar{a}-a\|_2^2 + \gamma Q^\pi(s',\bar{a})\right]dads' \tag{19}$$

$$= \int_{\mathcal{A}} p_D(\bar{a}|s,a,\sigma)(R(s,a) - \|\bar{a}-a\|_2^2)da$$

$$\qquad\qquad + \gamma\int_{\mathcal{S}}\mathbb{E}_{\bar{a}\sim\pi(\cdot|s')}\left[Q^\pi(s',\bar{a})\right]\int_{\mathcal{A}} p_D(s'|s,a)p_D(\bar{a}|s,a,\sigma)dads' \tag{20}$$

$$= R_\sigma(s,\bar{a}) + \gamma\mathbb{E}_{s'\sim P_\sigma(\cdot|s,\bar{a}),\bar{a}\sim\pi(\cdot|s')}\left[Q^\pi(s',\bar{a})\right] \tag{21}$$

Therefore, $Q^*$ satisfies the Bellman equation in NAMDP and is $Q$-value of $\pi$ in NAMDP $\qquad\square$

This result enables us to analyze how the choice of noise distribution in PANI influences the learned $Q$-values, by comparing the ground-truth Q-function under the NAMDP to that of the original MDP.

## B.2 ERROR BOUND BETWEEN NAMDP AND THE TRUE MDP

In this subsection, we provide theoretical insight into the difference in expected returns between the Noisy Action MDP (NAMDP) and the original MDP. For this analysis, we assume that both the state space $\mathcal{S}$ and the action space $\mathcal{A}$ are finite, and that $R_\sigma$ is well-defined on $\mathcal{S}\times\mathcal{A}$.

**Theorem B.4** (Error Bound in NAMDP). *Let $\eta(\pi)$ and $\bar{\eta}(\pi)$ denote the expected returns in the true MDP $(\mathcal{S}, \mathcal{A}, R, P, \gamma)$ and the NAMDP $(\mathcal{S}, \mathcal{A}, R_\sigma, P_\sigma, \gamma)$, respectively. suppose that NAMDP reward function $R_\sigma$ is bounded, The error between them is bounded as:*

$$|\eta(\pi) - \bar{\eta}(\pi)| \le \epsilon_r + \epsilon_m,$$

*where:*

$$\epsilon_r = \mathbb{E}_{d^\pi}\left[\,|R(s,a) - R_\sigma(s,a)|\,\right],$$

$$\epsilon_m = 2\bar{r}_{max}\frac{\gamma\mathbb{E}_{d^\pi}\left[\mathbb{TV}(P(s'|s,a)\|P_\sigma(s'|s,a))\right]}{(1-\gamma)^2}.$$

*Here, $R(s,a)$, and $d^\pi$ are the reward functions and the discounted state-action visitation distribution in the true MDP, $\bar{r}_{max}$ is the maximum reward in NAMDP, $\gamma$ is the discount factor, $\mathbb{TV}$ is the total variation.*

*Proof.* Let $\bar{d}^{\pi}$ denote the discounted state-action visitation distribution induced by policy $\pi$ in the NAMDP. Then, the difference in expected returns between the true MDP and the NAMDP can be written as:

$$|\eta(\pi) - \bar{\eta}(\pi)| = \left| \sum_{s,a} \left( d^{\pi}(s,a)R(s,a) - \bar{d}^{\pi}(s,a)R_{\sigma}(s,a) \right) \right|.$$

Applying the triangle inequality, we decompose this expression as:

$$|\eta(\pi) - \bar{\eta}(\pi)| \leq \left| \sum_{s,a} d^{\pi}(s,a) \left( R(s,a) - R_{\sigma}(s,a) \right) \right| + \left| \sum_{s,a} \left( d^{\pi}(s,a) - \bar{d}^{\pi}(s,a) \right) R_{\sigma}(s,a) \right| \quad (22)$$

$$= \mathbb{E}_{(s,a) \sim d^{\pi}} \left[ |R(s,a) - R_{\sigma}(s,a)| \right] + \left| \sum_{s,a} \left( d^{\pi}(s,a) - \bar{d}^{\pi}(s,a) \right) R_{\sigma}(s,a) \right|. \quad (23)$$

For the second term, we apply Lemma A.1 from Lee et al. (2020), which yields:

$$\left| \sum_{s,a} \left( d^{\pi}(s,a) - \bar{d}^{\pi}(s,a) \right) R_{\sigma}(s,a) \right| \leq \frac{2\gamma \bar{r}_{\max}}{(1-\gamma)^2} \mathbb{E}_{(s,a) \sim d^{\pi}} \left[ \mathbb{TV}(P(s' \mid s,a) \parallel P_{\sigma}(s' \mid s,a)) \right],$$

where $\bar{r}_{\max} = \max_{s,a} |R_{\sigma}(s,a)|$.

Substituting this into the previous expression gives the final bound:

$$|\eta(\pi) - \bar{\eta}(\pi)| \leq \underbrace{\mathbb{E}_{d^{\pi}} \left[ |R(s,a) - R_{\sigma}(s,a)| \right]}_{\epsilon_r} + \underbrace{\frac{2\gamma \bar{r}_{\max}}{(1-\gamma)^2} \mathbb{E}_{d^{\pi}} \left[ \mathbb{TV}(P(s' \mid s,a) \parallel P_{\sigma}(s' \mid s,a)) \right]}_{\epsilon_m}.$$

$\square$

Therefore, if the noise distribution is reasonable, it assigns higher weight to dataset actions, where both the reward and transition dynamics of the true MDP are accurately represented. In this case, as long as the learned policy avoids OOD actions, the optimal policy in the NAMDP is expected to perform well in the original MDP. In the following subsections, we show that, in practice, policies optimized under appropriate noise settings in NAMDPs tend to avoid OOD actions.

### B.3 AVOIDING OOD ACTIONS IN NAMDP WHEN $\sigma \to 0$

In this subsection, we examine the behavior of the optimal policy in the NAMDP as the noise level $\sigma \to 0$. We show that under certain conditions, the limiting optimal policy avoids OOD actions; however, this result does not directly imply that the same behavior holds for all NAMDPs when $\sigma$ is merely small.

Instead, we establish an intermediate result in this subsection that will later allow us to prove, in the next subsection, that optimal policies in NAMDPs with a finite action space indeed avoid OOD actions under small noise.

We begin by stating an assumption on the noise distribution, which will be used throughout the analysis:

**Assumption B.5.** The noise distribution $q_{\sigma}(\bar{a} \mid a)$ satisfies the following properties for all $a, a_1, a_2 \in \mathcal{A}$:

1. If $\|a_1 - a\|_2^2 > \|a_2 - a\|_2^2$, then

$$\lim_{\sigma \to 0^+} \frac{q_{\sigma}(a \mid a_1)}{q_{\sigma}(a \mid a_2)} = 0.$$

2. If $\|a_1 - a\|_2^2 = \|a_2 - a\|_2^2$, then

$$\lim_{\sigma \to 0^+} \frac{q_\sigma(a \mid a_1)}{q_\sigma(a \mid a_2)} = 1.$$

For example, distributions such as the Gaussian and Laplace distributions satisfy the properties. Given an arbitrary function $S : \mathcal{A} \times \mathcal{A} \to \mathbb{R}$, the following lemma characterizes the limiting behavior of a noise-weighted average as $\sigma \to 0$.

**Lemma B.6.** *Given a noise distribution $q_\sigma$, if the support of $p$ is finite, then for any function $S$, we have:*

$$\lim_{\sigma \to 0^+} \int_{\mathcal{A}} S(a, \bar{a})p(a|\bar{a}) \, da = \frac{\sum_{a \in C(\bar{a},p)} S(a, \bar{a})p(a)}{\sum_{a \in C(\bar{a},p)} p(a)} = \mathbb{E}_{a \sim p_C(\cdot|\bar{a})} \left[ S(a, \bar{a}) \right], \qquad (24)$$

*where*

$$p(a|\bar{a}) = \frac{q_\sigma(\bar{a}|a)p(a)}{\sum_a q_\sigma(\bar{a}|a)p(a)}, C(\bar{a}, p) = \left\{ a \in supp(p) \mid \|\bar{a} - a\|_2^2 \le \|\bar{a} - \mathbf{y}\|_2^2 \forall \mathbf{y} \in supp(p) \right\}. \qquad (25)$$

*Proof.* We analyze the expectation by splitting it into two cases: points in $C(\bar{a}, p)$ and points outside $C(\bar{a}, p)$.

Let $a^* \in C(\bar{a}, p)$ be a closest point to $\bar{a}$. By the Assumption B.5, $\frac{q_\sigma(\bar{a}|a)}{q_\sigma(\bar{a}|a^*)} \to 0$ as $\sigma \to 0^+$ for $a \notin C(\bar{a}, p)$. Thus:

$$\lim_{\sigma \to 0^+} \sum_{a \notin C(\bar{a},p)} S(a, \bar{a}) \frac{q_\sigma(\bar{a}|a)p(a)}{\sum_a q_\sigma(\bar{a}|a)p(a)} \qquad (26)$$

$$= \lim_{\sigma \to 0^+} \frac{\sum_{a \notin C(\bar{a},p)} S(a, \bar{a})q_\sigma(\bar{a}|a)p(a)}{\sum_{a \in C(\bar{a},p)} q_\sigma(\bar{a}|a)p(a) + \sum_{a \notin C(\bar{a},p)} q_\sigma(\bar{a}|a)p(a)}$$

$$= \lim_{\sigma \to 0^+} \frac{\sum_{a \notin C(\bar{a},p)} S(a, \bar{a}) \frac{q_\sigma(\bar{a}|a)}{q_\sigma(\bar{a}|a^*)} p(a)}{\sum_{a \in C(\bar{a},p)} \frac{q_\sigma(\bar{a}|a)}{q_\sigma(\bar{a}|a^*)} p(a) + \sum_{a \notin C(\bar{a},p)} \frac{q_\sigma(\bar{a}|a)}{q_\sigma(\bar{a}|a^*)} p(a)}$$

$$= \frac{0}{\sum_{a \in C(\bar{a},p)} p(a) + 0} = 0. \qquad (27)$$

For $a \in C(\bar{a}, p)$, the noise property gives $\frac{q_\sigma(\bar{a}|a)}{q_\sigma(\bar{a}|\bar{a})} \to 1$ for all $a, \bar{a} \in C(\bar{a}, p)$. Hence:

$$\lim_{\sigma \to 0^+} \sum_{a \in C(\bar{a},p)} S(a, \bar{a}) \frac{q_\sigma(\bar{a}|a)p(a)}{\sum_a q_\sigma(\bar{a}|a)p(a)} \qquad (28)$$

$$= \lim_{\sigma \to 0^+} \frac{\sum_{a \in C(\bar{a},p)} S(a, \bar{a})q_\sigma(\bar{a}|a)p(a)}{\sum_{a \in C(\bar{a},p)} q_\sigma(\bar{a}|a)p(a) + \sum_{a \notin C(\bar{a},p)} q_\sigma(\bar{a}|a)p(a)}$$

$$= \lim_{\sigma \to 0^+} \frac{\sum_{a \in C(\bar{a},p)} S(a, \bar{a}) \frac{q_\sigma(\bar{a}|a)}{q_\sigma(\bar{a}|a^*)} p(a)}{\sum_{a \in C(\bar{a},p)} \frac{q_\sigma(\bar{a}|a)}{q_\sigma(\bar{a}|a^*)} p(a) + \sum_{a \notin C(\bar{a},p)} \frac{q_\sigma(\bar{a}|a)}{q_\sigma(\bar{a}|a^*)} p(a)}$$

$$= \frac{\sum_{a \in C(\bar{a},p)} S(a, \bar{a})p(a)}{\sum_{a \in C(\bar{a},p)} p(a)}. \qquad (29)$$

Therefore, we have

$$\lim_{\sigma \to 0^+} \int_a S(a, \bar{a})p(a|\bar{a}) \, da = \lim_{\sigma \to 0^+} \left( \sum_{a \in C(\bar{a},p)} S(a, \bar{a})p(a|\bar{a}) + \sum_{a \notin C(\bar{a},p)} S(a, \bar{a})p(a|\bar{a}) \right)$$

$$= \frac{\sum_{a \in C(\bar{a},p)} S(a, \bar{a})p(a)}{\sum_{a \in C(\bar{a},p)} p(a)} = \mathbb{E}_{a \sim p_C(\cdot|\bar{a})} \left[ S(a, \bar{a}) \right], \qquad (30)$$

where $p_C(a|\bar{a}) = \frac{p(a)}{\sum_{a \in C(\bar{a},p)} p(a)}$ is the restriction of $p$ to $C(\bar{a},p)$. $\qquad \square$

Using this lemma, we can express the $Q$-value in closed form as $\sigma \to 0^+$, as shown below:

**Lemma B.7.** *suppose that $\sigma \to 0^+$, then the Q value in NAMDP holds:*

$$\mathbb{E}_{a \sim p_C(\cdot|\bar{a},s)}[Q^\pi(s,a)] = Q^\pi(s,\bar{a}) + \inf_{a \in C(\bar{a},p_D(\cdot|s))} \|\bar{a} - a\|_2^2.$$

*Proof.* We assume $\sigma \to 0^+$, leveraging the fact that the dataset is finite. By Lemma B.6, the transition probabilities in the NAMDP converge as:

$$\bar{P}(s'|s,\bar{a}) = \lim_{\sigma \to 0^+} \int p_D(s'|s,a)p_D(\bar{a}|s,a,\sigma)\,da \tag{31}$$

$$= \frac{\sum_{a \in C(\bar{a},p_D(\cdot|s))} p_D(s'|s,a)p_D(a|s)}{\sum_{a \in C(\bar{a},p_D(\cdot|s))} p_D(a|s)} \tag{32}$$

$$= \mathbb{E}_{a \sim p_C(\cdot|\bar{a},s)}[p_D(s'|s,a)]. \tag{33}$$

For the reward $\bar{R}(s,\bar{a})$, we derive:

$$\bar{R}(s,\bar{a}) = \lim_{\sigma \to 0^+} \int_{\mathcal{A}} p_D(\bar{a}|s,a,\sigma)(R(s,a) - \|a - \bar{a}\|_2^2)\,da \tag{34}$$

$$= \frac{\sum_{a \in C(\bar{a},p_D(\cdot|s))} p_D(a|s)(R(s,a) - \|a - \bar{a}\|_2^2)}{\sum_{a \in C(\bar{a},p_D(\cdot|s))} p_D(a|s)} \tag{35}$$

$$= \mathbb{E}_{a \sim p_C(\cdot|\bar{a},s)}[R(s,a) - \|a - \bar{a}\|_2^2]. \tag{36}$$

Next, note that the definition of $C(\bar{a}, p_D(\cdot|s))$ is:

$$C(\bar{a}, p_D(\cdot|s)) = \{a \in \operatorname{supp}(p_D(\cdot|s)) \mid \|\bar{a} - a\|_2^2 \le \|\bar{a} - b\|_2^2 \quad \forall b \in \operatorname{supp}(p_D(\cdot|s))\}.$$

This means $C(\bar{a}, p_D(\cdot|s))$ contains actions $a$ in $\operatorname{supp}(p_D(\cdot|s))$ that are closest to $\bar{a}$ in $L_2$-norm. By construction, the distribution $p_C(\cdot|\bar{a},s)$ assigns full probability mass to $C(\bar{a}, p_D(\cdot|s))$, and therefore:

$$\mathbb{E}_{a \sim p_C(\cdot|\bar{a},s)}[\|\bar{a} - a\|_2^2] = \inf_{a \in \operatorname{supp}(p_D(\cdot|s))} \|\bar{a} - a\|_2^2.$$

Now, using the Bellman equation for the NAMDP:

$$Q^\pi(s,\bar{a}) = \bar{R}(s,\bar{a}) + \gamma \mathbb{E}_{s' \sim \bar{p}(\cdot|s,\bar{a})}[V^\pi(s')].$$

Substituting $\bar{R}(s,a)$ and $\bar{P}(s'|s,a)$:

$$Q^\pi(s,\bar{a}) = \mathbb{E}_{a \sim p_C(\cdot|\bar{a},s)}\left[R(s,a) - \|\bar{a} - a\|_2^2\right] + \gamma \int_{\mathcal{S}} \mathbb{E}_{a \sim p_C(\cdot|\bar{a},s)}[p_D(s'|s,a)V^\pi(s')]\,ds' \tag{37}$$

$$= \mathbb{E}_{a \sim p_C(\cdot|\bar{a},s)}\left[R(s,a) - \|\bar{a} - a\|_2^2 + \gamma \mathbb{E}_{s' \sim p_D(\cdot|s,a)}[V^\pi(s')]\right] \tag{38}$$

$$= \mathbb{E}_{a \sim p_C(\cdot|\bar{a},s)}\left[R(s,a) + \gamma \mathbb{E}_{s' \sim p_D(\cdot|s,a)}[V^\pi(s')]\right] - \mathbb{E}_{a \sim p_C(\cdot|\bar{a},s)}\left[\|\bar{a} - a\|_2^2\right]. \tag{39}$$

$$= \mathbb{E}_{a \sim p_C(\cdot|\bar{a},s)}\left[\bar{R}(s,a) + \gamma \mathbb{E}_{s' \sim \bar{P}(\cdot|s,a)}[V^\pi(s')]\right] - \mathbb{E}_{a \sim p_C(\cdot|\bar{a},s)}\left[\|\bar{a} - a\|_2^2\right]. \tag{40}$$

$$= \mathbb{E}_{a \sim p_C(\cdot|\bar{a},s)}[Q^\pi(s,a)] - \mathbb{E}_{a \sim p_C(\cdot|\bar{a},s)}\left[\|\bar{a} - a\|_2^2\right]. \tag{41}$$

Finally, since $\mathbb{E}_{a \sim p_C(\cdot|\bar{a},s)}\left[[\|\bar{a} - a\|_2^2]\right] = \inf_{a \in \operatorname{supp}(p_D(\cdot|s))} \|\bar{a} - a\|_2^2$, we conclude:

$$\mathbb{E}_{a \sim p_C(\cdot|\bar{a},s)}[Q^\pi(s,a)] = Q^\pi(s,\bar{a}) + \inf_{a \in C(\bar{a},p_D(\cdot|s))} \|\bar{a} - a\|_2^2.$$

This completes the proof. $\qquad \square$

Using the above lemma, we can now show that the optimal policy avoids OOD actions in the limiting case, as formalized in the following corollary:

**Corollary B.8** (No OOD when $\sigma \to 0$). *Given a noise distribution $q_\sigma$, when $\sigma \to 0$, the optimal policy in the NAMDP is guaranteed to select actions within the dataset distribution.*

*Proof.* Suppose that $\pi^*$ is the optimal policy in the NAMDP when $\sigma \to 0$. Assume for contradiction that there exists $s \in \mathcal{S}$ and $\bar{a} \in \mathcal{A}$ such that $\bar{a} \in \operatorname{supp}(\pi^*(\cdot|s))$ but $\bar{a} \notin \operatorname{supp}(p_D(\cdot|s))$. This assumption implies that the policy selects an action $a$ outside the dataset distribution $p_D(\cdot|s)$ despite being optimal.

By lemma B.7, the $Q$-value can be expressed as:

$$\mathbb{E}_{a \sim p_C(\cdot|\bar{a},s)}\left[Q^*(s,a)\right] = Q^*(s,\bar{a}) + \inf_{a \in \operatorname{supp}(p_D(\cdot|s))} \|\bar{a} - a\|_2^2,$$

where the term $\inf_{a \in \operatorname{supp}(p_D(\cdot|s))} \|\bar{a} - a\|_2^2$ represents the minimum penalty induced by the distance between $\bar{a}$ and the closest action in $\operatorname{supp}(p_D(\cdot|s))$.

Since $\bar{a} \notin \operatorname{supp}(p_D(\cdot|s))$, this penalty term is strictly positive. Hence, substituting this into the inequality, we have:

$$\max_{a \in C(\bar{a}, p_D(\cdot|s))} Q^*(s,a) \geq \mathbb{E}_{a \sim p_C(\cdot|\bar{a},s)}\left[Q^*(s,a)\right] \tag{42}$$

$$= Q^*(s,\bar{a}) + \inf_{a \in \operatorname{supp}(p_D(\cdot|s))} \|\bar{a} - a\|_2^2 \tag{43}$$

$$> Q^*(s,\bar{a}). \tag{44}$$

This inequality implies that there exists another action $a \in C(\bar{a}, p_D(\cdot|s))$ (where $C(\bar{a}, p_D(\cdot|s))$ represents the set of nearby actions under the dataset distribution) with a higher $Q$-value than $Q^*(s,\bar{a})$. Thus, $\bar{a}$ cannot be optimal, contradicting the assumption that $\bar{a} \in \operatorname{supp}(\pi^*(\cdot|s))$. $\square$

## B.4 AVOIDING OOD ACTIONS UNDER SMALL NOISE

In this subsection, we further analyze the case where the action and state spaces are finite and show that, under sufficiently small noise, the optimal policy in the NAMDP avoids OOD actions. Specifically, we show that for any $\epsilon > 0$, there exists a $\sigma' > 0$ such that for all $0 < \sigma < \sigma'$, the Bellman optimality operator of the NAMDP is within $\epsilon$ of that of the noiseless case when $\sigma \to 0$.

Unless otherwise specified, we denote the NAMDP in the limit when $\sigma \to 0$ by $\bar{\mathcal{M}} = (\mathcal{S}, \mathcal{A}, \bar{R}, \bar{P}, \gamma)$, and the NAMDP with noise scale $\sigma$ by $\mathcal{M}_\sigma = (\mathcal{S}, \mathcal{A}, R_\sigma, P_\sigma, \gamma)$.

**Lemma B.9.** *Let $\bar{\mathcal{T}}$ denote the Bellman optimality operator of the NAMDP $\bar{\mathcal{M}}$, and let $\mathcal{T}_\sigma$ denote the Bellman optimality operator of the NAMDP $\mathcal{M}_\sigma$. Let $Q$ denote the $Q$-function under $\bar{\mathcal{M}}$. Then, for any $\epsilon > 0$, there exists $\sigma' > 0$ such that for all $0 < \sigma < \sigma'$,*

$$\sup_{(s,a) \in \mathcal{S} \times \mathcal{A}} \left|(\bar{\mathcal{T}}Q)(s,a) - (\mathcal{T}_\sigma Q)(s,a)\right| < \epsilon.$$

*Proof.* Let $\bar{\mathcal{T}}$ and $\mathcal{T}_\sigma$ be defined as in Lemma B.9. Fix any $\epsilon > 0$. Since $R_\sigma \to \bar{R}$ and $P_\sigma \to \bar{P}$ as $\sigma \to 0$, for each $(s,a)$, there exists $\sigma^R_{(s,a)} > 0$ such that for all $0 < \sigma < \sigma^R_{(s,a)}$,

$$|\bar{R}(s,a) - R_\sigma(s,a)| < \frac{\epsilon}{2}.$$

Similarly, for each $(s,a,s')$, there exists $\sigma^P_{(s,a,s')} > 0$ such that for all $0 < \sigma < \sigma^P_{(s,a,s')}$,

$$|\bar{P}(s'|s,a) - P_\sigma(s'|s,a)| < \frac{(1-\gamma)\epsilon}{2\gamma|\mathcal{S}|\max_{s,a}|\bar{R}(s,a)|}.$$

We now bound the difference between the Bellman operators, for any $(s,a)$,

$$|(\bar{\mathcal{T}}Q)(s,a) - (\mathcal{T}_\sigma Q)(s,a)| \tag{45}$$

$$\leq |\bar{R}(s,a) - R_\sigma(s,a)| + \gamma \left| \sum_{s'} \left( \bar{P}(s'|s,a) - P_\sigma(s'|s,a) \right) \max_{\bar{a}} Q(s',\bar{a}) \right| \tag{46}$$

$$\leq |\bar{R}(s,a) - R_\sigma(s,a)| + \gamma|\mathcal{S}| \max_{(s,a,s')} |\bar{P}(s'|s,a) - P_\sigma(s'|s,a)| \max_{(s,a)} |Q(s,a)| \tag{47}$$

$$\leq |\bar{R}(s,a) - R_\sigma(s,a)| + \frac{\gamma|\mathcal{S}| \max_{(s,a,s')} |\bar{P}(s'|s,a) - P_\sigma(s'|s,a)| \max_{(s,a)} |\bar{R}(s,a)|}{1-\gamma}. \tag{48}$$

Define $\sigma' = \min \left( \{\sigma^R_{(s,a)}\}_{(s,a)} \cup \{\sigma^P_{(s,a,s')}\}_{(s,a,s')} \right)$. Since the state and action spaces are finite, this minimum is taken over a finite set, ensuring that $\sigma' > 0$.

Then, for all $0 < \sigma < \sigma'$, we have

$$\sup_{(s,a)\in\mathcal{S}\times\mathcal{A}} |(\bar{\mathcal{T}}Q)(s,a) - (\mathcal{T}_\sigma Q)(s,a)| \tag{49}$$

$$\leq \sup_{(s,a)\in\mathcal{S}\times\mathcal{A}} |\bar{R}(s,a) - R_\sigma(s,a)| \tag{50}$$

$$+ \frac{\gamma|\mathcal{S}| \max_{(s,a,s')} |\bar{P}(s'|s,a) - P_\sigma(s'|s,a)| \max_{(s,a)} |\bar{R}(s,a)|}{1-\gamma} \tag{51}$$

$$< \frac{\epsilon}{2} + \frac{\epsilon}{2} = \epsilon, \tag{52}$$

completing the proof. $\qquad\square$

As a direct consequence of Lemma B.9, we obtain the following corollary:

**Corollary B.10.** *Let $Q^*$ denote the optimal $Q$-function of $\bar{\mathcal{M}}$, and let $Q^*_\sigma$ denote the optimal $Q$-function of $\mathcal{M}_\sigma$. Then, for any $\epsilon > 0$, there exists $\sigma' > 0$ such that for all $0 < \sigma < \sigma'$,*

$$\sup_{(s,a)\in\mathcal{S}\times\mathcal{A}} |Q^*(s,a) - Q^*_\sigma(s,a)| < \epsilon.$$

*Proof.* Fix any $\epsilon > 0$. By Lemma B.9, there exists $\sigma' > 0$ such that for all $0 < \sigma < \sigma'$,

$$\sup_{(s,a)\in\mathcal{S}\times\mathcal{A}} \left| (\bar{\mathcal{T}}Q^*)(s,a) - (\mathcal{T}_\sigma Q^*)(s,a) \right| < (1-\gamma)\epsilon.$$

Since $Q^*$ and $Q^*_\sigma$ are the fixed points of $\bar{\mathcal{T}}$ and $\mathcal{T}_\sigma$, respectively, and since the Bellman optimality operator is a $\gamma$-contraction, we have:

$$\sup_{(s,a)\in\mathcal{S}\times\mathcal{A}} |Q^*(s,a) - Q^*_\sigma(s,a)| \tag{53}$$

$$= \sup_{(s,a)\in\mathcal{S}\times\mathcal{A}} \left| (\bar{\mathcal{T}}Q^*)(s,a) - (\mathcal{T}_\sigma Q^*_\sigma)(s,a) \right| \tag{54}$$

$$\leq \sup_{(s,a)} \left| (\bar{\mathcal{T}}Q^*)(s,a) - (\mathcal{T}_\sigma Q^*)(s,a) \right| + \sup_{(s,a)} |(\mathcal{T}_\sigma Q^*)(s,a) - (\mathcal{T}_\sigma Q^*_\sigma)(s,a)| \tag{55}$$

$$< (1-\gamma)\epsilon + \gamma \sup_{(s,a)} |Q^*(s,a) - Q^*_\sigma(s,a)|. \tag{56}$$

Rearranging gives:

$$\sup_{(s,a)\in\mathcal{S}\times\mathcal{A}} |Q^*(s,a) - Q^*_\sigma(s,a)| < \frac{(1-\gamma)\epsilon}{1-\gamma} = \epsilon. \tag{57}$$

$\qquad\square$

Using Lemma B.7 and Corollary B.10, we can now show that the optimal policy in $\mathcal{M}_\sigma$ selects actions close to those in the dataset for sufficiently small noise.

**Theorem B.11** (No OOD Action Selection). *Let $\pi_\sigma^*$ be the optimal policy in the NAMDP $\mathcal{M}_\sigma$. Then, for any $\epsilon > 0$, there exists $\sigma' > 0$ such that for all $0 < \sigma < \sigma'$ and for all $\bar{a} \in supp(\pi_\sigma^*(\cdot|s))$,*

$$\inf_{a \in supp(p_D(\cdot|s))} \|\bar{a} - a\|_2^2 < \epsilon,$$

*where $supp(p_D(\cdot|s))$ denotes the support of the behavior policy $p_D$, that is, the set of actions observed in the dataset at state $s$.*

*Proof.* Assume for contradiction that there exists $\epsilon' > 0$ such that for all $\sigma' > 0$, there exists $0 < \sigma < \sigma'$ and $\bar{a} \in supp(\pi_\sigma^*(\cdot|s))$ satisfying

$$\inf_{a \in supp(p_D(\cdot|s))} \|\bar{a} - a\|_2^2 \geq \epsilon'.$$

By Corollary B.10, there exists $\sigma' > 0$ such that for all $0 < \sigma < \sigma'$,

$$\sup_{(s,a) \in \mathcal{S} \times \mathcal{A}} |Q^*(s,a) - Q_\sigma^*(s,a)| < \frac{\epsilon'}{4}. \tag{58}$$

Next, using Lemma B.7, which states that

$$\mathbb{E}_{a \sim p_C(\cdot|\bar{a},s)}[Q^*(s,a)] = Q^*(s,\bar{a}) + \inf_{a \in C(\bar{a},p_D(\cdot|s))} \|\bar{a} - a\|_2^2,$$

we proceed as follows:

First, observe that the expected $Q$-value over $C(\bar{a}, p_D(\cdot|s))$ under $Q_\sigma^*$ is lower bounded by the same expectation under $Q^*$, up to the error $\epsilon'/4$ from Eq. equation 58:

$$\max_{a \in C(\bar{a},p_D(\cdot|s))} Q_\sigma^*(s,a) \geq \mathbb{E}_{a \sim p_D(\cdot|s)}[Q_\sigma^*(s,a)] \tag{59}$$

$$> \mathbb{E}_{a \sim p_D(\cdot|s)}[Q^*(s,a)] - \frac{\epsilon'}{4}. \tag{60}$$

Applying Lemma B.7, we can expand this expectation as:

$$\mathbb{E}_{a \sim p_D(\cdot|s)}[Q^*(s,a)] = Q^*(s,\bar{a}) + \inf_{a \in supp(p_D(\cdot|s))} \|\bar{a} - a\|_2^2. \tag{61}$$

Substituting this into the inequality above yields:

$$\max_{a \in C(\bar{a},p_D(\cdot|s))} Q_\sigma^*(s,a) > Q^*(s,\bar{a}) + \inf_{a \in supp(p_D(\cdot|s))} \|\bar{a} - a\|_2^2 - \frac{\epsilon'}{4} \tag{62}$$

$$> Q_\sigma^*(s,\bar{a}) + \inf_{a \in supp(p_D(\cdot|s))} \|\bar{a} - a\|_2^2 - \frac{\epsilon'}{2}, \tag{63}$$

where the second inequality uses the fact that $Q^*$ and $Q_\sigma^*$ are close by at most $\epsilon'/4$ in both directions.

Since we have assumed that $\inf_{a \in supp(p_D(\cdot|s))} \|\bar{a} - a\|_2^2 \geq \epsilon'$, this further implies:

$$\max_{a \in C(\bar{a},p_D(\cdot|s))} Q_\sigma^*(s,a) \geq Q_\sigma^*(s,\bar{a}) + \frac{\epsilon'}{2} > Q_\sigma^*(s,\bar{a}) \tag{64}$$

This shows that there exists an action in $C(\bar{a}, p_D(\cdot|s))$ that achieves strictly higher $Q$-value than $\bar{a}$, meaning $\bar{a}$ cannot be optimal. This contradicts the assumption that $Q_\sigma^*$ is the optimal $Q$-value.

$\square$

## C  IMPLEMENTATION DETAILS

All experiments were conducted on a single NVIDIA RTX 3090 GPU with an Intel Xeon Gold 6330 CPU.

Our code is available at: https://anonymous.4open.science/r/PANI-1EED

### C.1  IQL-AN

---

**Algorithm 1** IQL with Penalized Action Noise Injection (IQL-AN)

---

Initialize critic networks $Q_{\theta_1}$, $Q_{\theta_2}$, value network $V_\psi$ and actor network $\pi_\phi$ with random parameters $\theta_1$, $\theta_2$, $\psi$ and $\phi$
Initialize target networks: $\theta_1' \leftarrow \theta_1$, $\theta_2' \leftarrow \theta_2$, $\phi' \leftarrow \phi$
**for** $t = 1$ to $T$ **do**
    Sample a mini-batch of transitions $(s, a, r, s')$ from the dataset $\mathcal{D}$
    Sample a noisy action $\bar{a}$ from the distribution $q_\sigma(\cdot \mid a)$
    **Value update:**
    Minimize the PANI value objective $J_V^{\text{IQL}}(\phi)$ equation 65
    **Critic update:**
    Minimize the PANI critic objective $J_Q^{\text{IQL}}(\theta)$ equation 66
    **Actor update:**
    **if** policy is deterministic **then**
        Minimize the PANI deterministic actor objective $J_{\det}^{\text{IQL}}(\phi)$ equation 67
    **else**
        Minimize the PANI stochastic actor objective $J_{\text{sto}}^{\text{IQL}}(\phi)$ equation 68
    **end if**
    **Target network update:**
    Update critic target networks: $\theta_i' \leftarrow \eta\theta_i + (1 - \eta)\theta_i'$
    Update actor target network: $\phi' \leftarrow \eta\phi + (1 - \eta)\phi'$
**end for**

---

IQL-AN extends IQL (Kostrikov et al., 2021) by incorporating Penalized Action Noise Injection, enhancing its ability to address OOD overestimation as detailed in Algorithm 1. The training process follows the objective functions defined below:

$$J_V^{\text{IQL}}(\psi) = \mathbb{E}_{(s,a)\sim\mathcal{D}} \left[ L_2^\tau \left( \min_{i=1,2} Q_{\theta_i'}(s, a) - V_\psi(s) \right) \right] \tag{65}$$

$$\text{where} \quad L_2^\tau(x) = |\tau - \mathbb{I}(u < 0)| x^2$$

$$J_Q^{\text{IQL}}(\theta) = \mathbb{E}_{\substack{(s,a,s')\sim\mathcal{D} \\ \bar{a}\sim q_\sigma(\cdot|a)}} \left[ \left( Q_\theta(s, \bar{a}) - (r(s, a) - \|a - \bar{a}\|_2^2 + \gamma V_\psi(s'))\right)^2 \right] \tag{66}$$

$$J_{\det}^{\text{IQL}}(\phi) = -\mathbb{E}_{(s,a)\sim\mathcal{D}} \left[ \min_{i=1,2} Q_{\theta_i'}(s, \pi_\phi(s)) \right] \tag{67}$$

$$J_{\text{sto}}^{\text{IQL}}(\phi) = -\mathbb{E}_{\substack{(s,a)\sim\mathcal{D} \\ \bar{a}\sim\pi_\phi(\cdot|s)}} \left[ \min_{i=1,2} Q_{\theta_i'}(s, \bar{a}) - \alpha \log \pi_\phi(a|s) \right] \tag{68}$$

In the Gym-MuJoCo environments, we used a deterministic policy, whereas in the AntMaze environments, we employed a unimodal Gaussian policy transformed via a hyperbolic tangent bijection. Additionally, we incorporated the NLL loss used in DTQL (Chen et al., 2024).

During training, experiments were conducted with $\log \sigma$ values of $-1, -5, -10, -20$ for Gym-MuJoCo. For $\tau$, we primarily set $\tau = 0.7$ across all Gym-MuJoCo tasks but additionally explored

$\tau = 0.99$ for the halfcheetah-medium-expert and halfcheetah-expert datasets. In the AntMaze environments, we tested with $\log \sigma$ values of $-10, -20$ and $\alpha$ values of $0.3, 0.5, 1.0$, while fixing $\tau = 0.9$.

The optimal hyperparameters for each environment are provided in Table 8, while the performance across different parameter settings is presented in Appendix C.5. For training curves, refer to Appendix C.6.

## C.2 TD3-AN

---

**Algorithm 2** TD3 with Penalized Action Noise Injection (TD3-AN)

---

Initialize critic networks $Q_{\theta_1}$, $Q_{\theta_2}$, and actor network $\pi_\phi$ with random parameters $\theta_1$, $\theta_2$, and $\phi$
Initialize target networks: $\theta'_1 \leftarrow \theta_1, \theta'_2 \leftarrow \theta_2, \phi' \leftarrow \phi$
**for** $t = 1$ to $T$ **do**
  Sample a mini-batch of transitions $(s, a, r, s')$ from the dataset $\mathcal{D}$
  Sample a noisy action $\bar{a}$ from the distribution $q_\sigma(\cdot \mid a)$
  **Critic update:**
  Compute the target action: $\tilde{a} \leftarrow \pi_{\phi'}(s') + \epsilon$, where $\epsilon \sim \text{clip}(\mathcal{N}(0, \bar{\sigma}), -c, c)$
  Minimize the PANI critic objective $J_Q^{\text{TD3}}(\theta)$ (69)
  **if** $t \bmod d = 0$ **then**
    **Actor update:**
    Minimize the PANI actor objective $J_\pi^{\text{TD3}}(\phi)$ (70)
    **Target network update:**
    Update critic target networks: $\theta'_i \leftarrow \eta\theta_i + (1 - \eta)\theta'_i$
    Update actor target network: $\phi' \leftarrow \eta\phi + (1 - \eta)\phi'$
  **end if**
**end for**

---

TD3-AN is an algorithm that applies Penalized Action Noise Injection to TD3 (Fujimoto et al., 2018). As described in Section 4, it can be implemented with only minor modifications, as shown in Algorithm 2. Specifically, it is trained using the following objective functions:

$$J_Q^{\text{TD3}}(\theta) = \mathbb{E}_{\substack{(s,a,s')\sim\mathcal{D} \\ \bar{a}\sim q_\sigma(\cdot|a)}} \left[ \left( Q_\theta(s, \bar{a}) - (r(s,a) - \|a - \bar{a}\|_2^2 + \gamma \min_{i=1,2} Q_{\theta'_i}(s', \tilde{a}) \right)^2 \right] \quad (69)$$

$$J_\pi^{\text{TD3}}(\phi) = -\mathbb{E}_{(s,a)\sim\mathcal{D}} \left[ \min_{i=1,2} Q_{\theta'_i}(s, \pi_\phi(s)) - \alpha\|a - \pi_\phi(s)\|_2^2 \right] \quad (70)$$

When training TD3-AN, we use the hybrid noise distribution described in Section 6.1 as the noise distribution. For the Gym-MuJoCo environments, experiments were conducted with $\alpha = 0$ and $\log \sigma$ values of $-1, -5, -10, -20$. In the AntMaze environments, we tested with $\alpha = 0.3, 0.5, 1.0$ and $\log \sigma$ values of $-5, -10, -20$.

The optimal hyperparameters for each environment are provided in Table 8, while the performance across different parameter settings is presented in Appendix C.5. For training curves, refer to Appendix C.6.

## C.3 QGPO-AN

In addition to TD3 and IQL, we further evaluated PANI on QGPO(Lu et al., 2023), a state-of-the-art diffusion-based algorithm. PANI can be integrated into QGPO with only minor modifications, as shown in Algorithm 3. For fair comparison, we used the same hyperparameters $(\beta_Q, K)$ as in the original QGPO. We also tested additional settings with guidance scales of $1.0, 2.0, 3.0, 5.0, 8.0$, and $10.0$, and $\log \sigma$ values of $-0.5, -1.0, -20.0$, and $-30.0$.

Due to computational constraints—each experiment requiring approximately 9 hours to complete—we were unable to perform extensive hyperparameter tuning. Nevertheless, the positive results obtained

with default parameters are encouraging and suggest that further performance gains may be possible with more thorough optimization. The optimal hyperparameters used for each environment are summarized in Table 6.

---

**Algorithm 3** QGPO with Penalized Action Noise Injection (QGPO-AN)

---

Initialize behavior model $\epsilon_\theta$, critic $Q_\psi$, energy model $f_\phi$
**Train behavior model:**
**for** each gradient step **do**
    Sample a mini-batch of transitions $(s, a) \sim \mathcal{D}$
    Sample noise $\epsilon \sim \mathcal{N}(0, I)$
    Sample time $t \sim \mathcal{U}(0, T)$
    Compute perturbed actions $a_t \leftarrow \alpha_t a + \sigma_t \epsilon$
    Minimize $\|\epsilon_\theta(a_t \mid s, t) - \epsilon\|_2^2$
**end for**
**Generate support actions:**
**for** each state $s$ in $\mathcal{D}^\mu$ **do**
    Sample $K$ actions $\hat{a}^{(1:K)} \sim \mu_\theta(\cdot \mid s)$
    Store $\hat{a}^{(1:K)}$ in $\mathcal{D}^{\mu_\theta}(s)$
**end for**
**Train critic and energy model:**
**for** each gradient step **do**
    Sample a mini-batch of transitions $(s, a, r, s') \sim \mathcal{D}$
    Sample noise $\epsilon \sim \mathcal{N}(0, I)$
    Sample time $t \sim \mathcal{U}(0, T)$
    Retrieve support actions $\hat{a}^{(1:K)} \sim \mathcal{D}^{\mu_\theta}(s)$ and $\hat{a}'^{(1:K)} \sim \mathcal{D}^{\mu_\theta}(s')$
    Compute target $Q$-value:

$$\mathcal{T}^\pi Q_\psi(s, a) = r + \gamma \sum_{\hat{a}'} \frac{\exp(\beta_Q Q_\psi(s', \hat{a}'))}{\sum_j \exp(\beta_Q Q_\psi(s', \hat{a}'_j))} Q_\psi(s', \hat{a}')$$

    Sample a noisy action $\bar{a}$ from the distribution $q_\sigma(\cdot \mid a)$
    Minimize $\|Q_\psi(s, \bar{a}) - \mathcal{T}^\pi Q_\psi(s, a) - \|a - \bar{a}\|_2^2\|_2^2$
    Compute perturbed support actions $\hat{a}_t \leftarrow \alpha_t \hat{a} + \sigma_t \epsilon$
    Maximize:
$$\sum_i \frac{\exp(\beta Q_\psi(s, \hat{a}_i))}{\sum_j \exp(\beta Q_\psi(s, \hat{a}_j))} \log \frac{\exp(f_\phi(\hat{a}_{i,t} \mid s, t))}{\sum_j \exp(f_\phi(\hat{a}_{j,t} \mid s, t))}$$

**end for**

---

## C.4 HYPERPARAMETERS

Table 4: TD3-AN's and IQL-AN's common hyperparameters

| Hyperparameter | Value |
|---|---|
| optimizer | Adam (Kingma, 2014) |
| batch size | 256 |
| learning rate (all networks) | $10^{-3}$ |
| target update rate ($\eta$) | $5 \times 10^{-3}$ |
| hidden dim (all networks) | 256, 512 (OGBench) |
| hidden layers (all networks) | 3, 4 (OGBench) |
| discount factor ($\gamma$) | 0.99 |
| training steps ($T$) | $10^6$, 20000 (adroit) |
| activation | ReLU |
| layer norm (Ba, 2016) (all networks) | True |

Table 5: TD3-AN's common hyperparameters

| Hyperparameter | Value |
|---|---|
| Policy update frequency ($d$) | 2 |
| Policy Noise ($\bar{\sigma}$) | 0.2 |
| Policy Noise Clipping ($c$) | 0.5 |

Table 6: Optimal hyperparameters for TD3-AN and IQL-AN across different environments

| Dataset | Environment | TD3 | | IQL | | |
|---|---|---|---|---|---|---|
| | | $\log \sigma$ | $\alpha$ | $\log \sigma$ | $\alpha$ | $\tau$ |
| medium | halfcheetah | -20.0 | - | -20.0 | - | 0.70 |
| medium | hopper | -5.0 | - | -5.0 | - | 0.70 |
| medium | walker2d | -5.0 | - | -20.0 | - | 0.70 |
| medium-replay | halfcheetah | -20.0 | - | -20.0 | - | 0.70 |
| medium-replay | hopper | -10.0 | - | -20.0 | - | 0.70 |
| medium-replay | walker2d | -5.0 | - | -5.0 | - | 0.70 |
| medium-expert | halfcheetah | -10.0 | - | -1.0 | - | 0.99 |
| medium-expert | hopper | -1.0 | - | -1.0 | - | 0.70 |
| medium-expert | walker2d | -20.0 | - | -10.0 | - | 0.70 |
| expert | halfcheetah | -20.0 | - | -1.0 | - | 0.99 |
| expert | hopper | -1.0 | - | -1.0 | - | 0.70 |
| expert | walker2d | -5.0 | - | -1.0 | - | 0.70 |
| full-replay | halfcheetah | -20.0 | - | -20.0 | - | 0.70 |
| full-replay | hopper | -20.0 | - | -5.0 | - | 0.70 |
| full-replay | walker2d | -10.0 | - | -10.0 | - | 0.70 |
| random | halfcheetah | -20.0 | - | -20.0 | - | 0.70 |
| random | hopper | -10.0 | - | -5.0 | - | 0.70 |
| random | walker2d | -10.0 | - | -10.0 | - | 0.70 |
| - | umaze | -20.0 | 1.0 | -10.0 | 1.0 | 0.90 |
| diverse | umaze | -20.0 | 1.0 | -20.0 | 1.0 | 0.90 |
| play | medium | -10.0 | 1.0 | -20.0 | 0.3 | 0.90 |
| diverse | medium | -10.0 | 1.0 | -10.0 | 0.3 | 0.90 |
| play | large | -20.0 | 0.5 | -20.0 | 0.3 | 0.90 |
| diverse | large | -10.0 | 1.0 | -20.0 | 0.3 | 0.90 |

Table 7: Hyperparameters used for Adroit and OGBench (antmaze-giant-navigate-singletask) experiments.

| Dataset | $\log \sigma$ | $\alpha$ |
|---|---|---|
| Adroit (pen-cloned) | -10 | 30 |
| Adroit (pen-human) | -10 | 100 |
| OGBench (task 0) | -20 | 1.0 |
| OGBench (task 1) | -20 | 1.0 |
| OGBench (task 3) | -20 | 1.0 |
| OGBench (task 4) | -20 | 0.5 |
| OGBench (task 5) | -20 | 1.0 |

Table 8: Optimal hyperparameters for QGPO-AN across different environments

| Dataset | Environment | $\log \sigma$ | guidance scale |
|---|---|---|---|
| medium | halfcheetah | -20.0 | 10.0 |
| medium | hopper | -20.0 | 10.0 |
| medium | walker2d | -30.0 | 10.0 |
| medium-replay | halfcheetah | -20.0 | 8.0 |
| medium-replay | hopper | -1.0 | 5.0 |
| medium-replay | walker2d | -0.5 | 8.0 |
| medium-expert | halfcheetah | -20.0 | 5.0 |
| medium-expert | hopper | -1.0 | 1.0 |
| medium-expert | walker2d | -20.0 | 10.0 |

## C.5 RESULTS

1436 1435 1434 1433 1432 1431 1430 1429 1428 1427 1426 1425 1424 1423 1422 1421 1420 1419 1418 1417 1416 1415 1414 1413 1412 1411 1410 1409 1408 1407 1406 1405 1404

27

Table 9: The average normalized scores, as suggested by D4RL, are reported from the final evaluation on Gym-MuJoCo and Antmaze tasks. For Gym-MuJoCo tasks, results are obtained using five independent training seeds and 10 trajectories per seed, while for Antmaze, 100 trajectories per seed are used. The $\pm$ symbol denotes the standard error of the mean performance across seeds. Performance metrics and standard errors for baseline methods are taken from their respective original papers, with the exception of IQL and TD3+BC, which are sourced from ReBRAC. Diffusion-QL and DTQL report their metrics differently - their $\pm$ values represent standard errors calculated across all trajectories from all seeds, rather than across the mean performances of individual seeds. The highest scores for each task are highlighted in bold, while the second-highest scores are underlined. The Average (medium) score includes only medium, medium-replay, and medium-expert datasets.

| Dataset | Environment | IQL | Diffusion-free | | SfBC | Diffusion-policy | | Diffusion-based | | Ours | |
| | | | TD3+BC | ReBRAC | | D-QL* | QGPO | SRPO | DTQL* | IQL-AN | TD3-AN |
|---|---|---|---|---|---|---|---|---|---|---|---|
| medium | halfcheetah | 50.0 ± 0.1 | 54.7 ± 0.3 | **65.6** ± 0.3 | 45.9 ± 0.7 | 51.1 ± 0.5 | 54.1 ± 0.2 | 60.4 ± 0.3 | 57.9 ± 0.1 | 55.4 ± 0.3 | 61.5 ± 0.3 |
| medium | hopper | 65.2 ± 1.3 | 60.9 ± 2.4 | **102.0** ± 0.3 | 57.1 ± 1.3 | 90.5 ± 4.6 | 98.0 ± 1.2 | 95.5 ± 0.8 | 99.6 ± 0.9 | 98.4 ± 1.2 | 98.2 ± 0.9 |
| medium | walker2d | 80.7 ± 1.1 | 77.7 ± 0.9 | 82.5 ± 1.1 | 77.9 ± 0.8 | 87.0 ± 0.9 | 86.0 ± 0.3 | 84.4 ± 1.8 | **89.4** ± 0.1 | 87.5 ± 3.7 | 88.5 ± 0.6 |
| medium-replay | halfcheetah | 42.1 ± 1.1 | 45.0 ± 0.3 | 51.0 ± 0.3 | 37.1 ± 0.5 | 47.8 ± 0.3 | 47.6 ± 0.6 | 51.4 ± 1.4 | 50.9 ± 0.1 | 49.5 ± 0.4 | **53.3** ± 0.3 |
| medium-replay | hopper | 89.6 ± 4.2 | 55.1 ± 10.0 | 98.1 ± 1.7 | 86.2 ± 2.9 | 100.7 ± 0.6 | 96.9 ± 1.2 | 101.2 ± 0.4 | 100.0 ± 0.1 | 100.8 ± 0.4 | **102.3** ± 0.2 |
| medium-replay | walker2d | 75.4 ± 2.9 | 68.0 ± 6.1 | 77.3 ± 2.5 | 65.1 ± 1.8 | **95.5** ± 1.5 | 84.4 ± 1.8 | 84.6 ± 2.9 | 88.5 ± 2.2 | 88.8 ± 3.6 | 87.8 ± 6.3 |
| medium-expert | halfcheetah | 92.7 ± 0.9 | 89.1 ± 1.8 | **101.1** ± 1.6 | 92.6 ± 0.2 | 96.8 ± 0.3 | 93.5 ± 0.1 | 92.2 ± 1.2 | 92.7 ± 0.2 | 89.9 ± 2.3 | 96.4 ± 0.8 |
| medium-expert | hopper | 85.5 ± 9.4 | 87.8 ± 3.3 | 107.0 ± 2.0 | 108.6 ± 0.7 | **111.1** ± 1.3 | 108.0 ± 1.1 | 100.1 ± 5.7 | 109.3 ± 1.5 | 105.3 ± 3.7 | 108.8 ± 0.9 |
| medium-expert | walker2d | 112.1 ± 0.2 | 110.4 ± 0.2 | 111.6 ± 0.1 | 109.8 ± 0.1 | 110.1 ± 0.3 | 110.7 ± 0.3 | **114.0** ± 0.9 | 110.0 ± 0.1 | 109.6 ± 0.6 | **114.9** ± 0.2 |
| expert | halfcheetah | 95.5 ± 0.7 | 93.4 ± 0.1 | **105.9** ± 0.5 | - | - | - | - | - | 93.8 ± 0.2 | 104.4 ± 3.8 |
| expert | hopper | 108.8 ± 1.0 | **109.6** ± 1.2 | 100.1 ± 2.6 | - | - | - | - | - | 108.6 ± 2.6 | 109.0 ± 3.0 |
| expert | walker2d | 96.9 ± 10.2 | 110.0 ± 0.2 | 112.3 ± 0.1 | - | - | - | - | - | 108.2 ± 0.2 | **112.8** ± 0.2 |
| full-replay | halfcheetah | 75.0 ± 0.2 | 75.0 ± 0.8 | **82.1** ± 0.3 | - | - | - | - | - | 78.3 ± 0.1 | 81.2 ± 0.5 |
| full-replay | hopper | 104.4 ± 3.4 | 97.9 ± 5.5 | 107.1 ± 0.1 | - | - | - | - | - | 105.9 ± 0.4 | **108.3** ± 0.1 |
| full-replay | walker2d | 97.5 ± 0.4 | 90.3 ± 1.7 | 102.2 ± 0.5 | - | - | - | - | - | 100.6 ± 1.7 | 103.8 ± 0.7 |
| random | halfcheetah | 19.5 ± 0.3 | **30.9** ± 0.1 | 29.5 ± 0.5 | - | - | - | - | - | 24.3 ± 3.1 | 30.0 ± 0.5 |
| random | hopper | **10.1** ± 1.9 | 8.5 ± 0.2 | 8.1 ± 0.8 | - | - | - | - | - | 9.0 ± 0.2 | 10.0 ± 0.6 |
| random | walker2d | 11.3 ± 2.2 | 2.0 ± 1.1 | 18.4 ± 1.4 | - | - | - | - | - | **19.4** ± 2.8 | 5.8 ± 0.7 |
| **Average (Gym-MuJoCo)** | | 72.9 | 70.3 | 81.2 | - | - | - | - | - | 79.6 | **82.1** |
| **Average (medium)** | | 77.0 | 72.1 | 88.5 | 75.6 | 88.0 | 86.6 | 87.1 | 88.7 | 87.2 | **90.2** |
| - | umaze | 83.3 ± 1.4 | 66.3 ± 2.0 | 97.8 ± 0.3 | 92.0 ± 0.7 | 93.4 ± 3.4 | 96.4 ± 0.6 | 97.1 ± 1.1 | 92.6 ± 1.2 | 91.2 ± 1.1 | **98.4** ± 0.5 |
| diverse | umaze | 70.6 ± 1.2 | 53.8 ± 2.7 | **88.3** ± 4.1 | 85.3 ± 1.1 | 66.2 ± 8.6 | 74.4 ± 4.3 | 82.1 ± 4.4 | 74.4 ± 1.9 | 68.0 ± 3.2 | 74.6 ± 4.9 |
| play | medium | 64.6 ± 1.5 | 26.5 ± 5.8 | **84.0** ± 1.3 | 81.3 ± 0.8 | 76.6 ± 10.8 | 83.6 ± 2.0 | 80.7 ± 2.9 | 76.0 ± 1.9 | 74.4 ± 3.7 | 83.8 ± 2.7 |
| diverse | medium | 61.7 ± 1.9 | 25.9 ± 4.8 | 76.3 ± 4.3 | 82.0 ± 1.0 | 78.6 ± 10.3 | 83.8 ± 1.6 | 75.0 ± 5.0 | 80.6 ± 1.8 | 75.2 ± 1.6 | **85.8** ± 1.2 |
| play | large | 42.5 ± 2.1 | 0.0 ± 0.0 | 60.4 ± 8.3 | 59.3 ± 4.5 | 46.4 ± 8.3 | **66.6** ± 4.4 | 53.6 ± 5.1 | 59.2 ± 2.2 | 49.4 ± 3.0 | 65.4 ± 2.7 |
| diverse | large | 27.6 ± 2.5 | 0.0 ± 0.0 | 54.4 ± 7.9 | 45.5 ± 2.1 | 56.6 ± 7.6 | **64.8** ± 2.5 | 53.6 ± 2.6 | 62.0 ± 2.2 | 52.8 ± 2.6 | 58.0 ± 5.1 |
| **Average (AntMaze)** | | 58.4 | 28.8 | 76.9 | 74.2 | 69.6 | **78.3** | 73.7 | 74.1 | 68.5 | 77.7 |

Table 10: Comparison of QGPO and QGPO+AN performance across different environments. The $\pm$ values represent the standard error of algorithm performance across 3 random seeds.

| Environment | QGPO | QGPO-AN |
|---|---|---|
| halfcheetah-medium-expert | 93.5 | $93.6 \pm 0.3$ (+0.1) |
| hopper-medium-expert | 108.0 | $111.2 \pm 1.7$ (+3.2) |
| walker2d-medium-expert | 110.7 | $111.0 \pm 0.4$ (+0.3) |
| halfcheetah-medium | 54.1 | $53.8 \pm 0.4$ (-0.3) |
| hopper-medium | 98.0 | $99.4 \pm 1.1$ (+1.4) |
| walker2d-medium | 86.0 | $86.1 \pm 0.4$ (+0.1) |
| halfcheetah-medium-replay | 47.6 | $47.6 \pm 0.1$ (-0.0) |
| hopper-medium-replay | 96.9 | $99.8 \pm 0.5$ (+2.9) |
| walker2d-medium-replay | 84.4 | $89.7 \pm 1.5$ (+5.3) |

Table 11: Final performance of TD3-AN with Gaussian Noise Distribution across different $\log \sigma$ values in Gym-MuJoCo environments. Results are averaged over five training seeds with 10 trajectories per seed. Standard deviations are indicated by $\pm$, and the highest scores within 5% of the best per task are highlighted in bold.

| Dataset | Environment | 0.0 | -0.5 | -1.0 | -2.0 |
|---|---|---|---|---|---|
| medium | halfcheetah | $4.70\pm6.79$ | $47.64\pm0.69$ | $51.76\pm0.41$ | $\mathbf{65.36}\pm1.38$ |
| medium | hopper | $47.91\pm2.24$ | $\mathbf{76.00}\pm2.60$ | $71.41\pm23.17$ | $3.98\pm3.90$ |
| medium | walker2d | $54.84\pm27.71$ | $\mathbf{82.77}\pm2.33$ | $\mathbf{85.67}\pm1.47$ | $0.29\pm0.82$ |
| medium-replay | halfcheetah | $8.48\pm8.63$ | $22.49\pm10.17$ | $\mathbf{35.76}\pm4.75$ | $34.69\pm3.66$ |
| medium-replay | hopper | $1.82\pm0.01$ | $40.63\pm38.92$ | $\mathbf{79.68}\pm14.52$ | $22.08\pm7.66$ |
| medium-replay | walker2d | $-0.22\pm0.06$ | $-0.21\pm0.02$ | $\mathbf{63.36}\pm27.89$ | $5.46\pm2.18$ |
| medium-expert | halfcheetah | $41.80\pm4.72$ | $84.29\pm4.21$ | $\mathbf{89.94}\pm4.55$ | $67.52\pm23.73$ |
| medium-expert | hopper | $58.54\pm26.48$ | $\mathbf{107.21}\pm8.34$ | $70.02\pm26.77$ | $1.52\pm0.49$ |
| medium-expert | walker2d | $78.38\pm41.07$ | $\mathbf{108.40}\pm0.37$ | $\mathbf{103.35}\pm11.21$ | $-0.12\pm0.04$ |
| expert | halfcheetah | $86.06\pm3.94$ | $\mathbf{93.10}\pm0.36$ | $\mathbf{97.69}\pm0.60$ | $4.20\pm4.43$ |
| expert | hopper | $\mathbf{104.92}\pm4.26$ | $\mathbf{108.02}\pm6.66$ | $76.50\pm18.32$ | $1.03\pm0.32$ |
| expert | walker2d | $\mathbf{108.33}\pm0.52$ | $\mathbf{108.99}\pm0.19$ | $\mathbf{109.87}\pm0.09$ | $18.99\pm23.86$ |
| full-replay | halfcheetah | $0.34\pm0.30$ | $69.54\pm1.47$ | $73.94\pm0.74$ | $\mathbf{80.42}\pm1.40$ |
| full-replay | hopper | $0.65\pm0.03$ | $\mathbf{105.52}\pm1.24$ | $\mathbf{108.56}\pm0.60$ | $39.47\pm12.38$ |
| full-replay | walker2d | $1.35\pm1.55$ | $71.36\pm43.82$ | $\mathbf{101.12}\pm1.79$ | $25.37\pm26.63$ |
| random | halfcheetah | $17.13\pm0.76$ | $22.26\pm1.08$ | $\mathbf{31.42}\pm0.75$ | $\mathbf{30.08}\pm2.54$ |
| random | hopper | $7.31\pm0.12$ | $7.59\pm0.14$ | $8.94\pm2.01$ | $\mathbf{9.77}\pm0.52$ |
| random | walker2d | $-0.09\pm0.01$ | $-0.10\pm0.00$ | $\mathbf{7.15}\pm0.59$ | $3.83\pm1.78$ |
| Average | | 34.57 | 64.20 | **70.34** | 23.00 |

Table 12: Final performance of TD3-AN with Laplace Noise Distribution across different $\log \sigma$ values in Gym-MuJoCo environments. Results are averaged over five training seeds with 10 trajectories per seed. Standard deviations are indicated by $\pm$, and the highest scores within 5% of the best per task are highlighted in bold.

| Dataset | Environment | 0.0 | -0.5 | -1.0 | -2.0 |
|---|---|---|---|---|---|
| medium | halfcheetah | 27.96±9.30 | 47.93±0.46 | 51.65±0.50 | **64.79±0.80** |
| medium | hopper | 53.43±5.83 | 84.00±10.10 | **93.21±3.67** | 6.79±5.63 |
| medium | walker2d | 74.03±7.65 | **84.62±0.50** | **84.97±1.80** | 0.50±1.15 |
| medium-replay | halfcheetah | 32.89±7.08 | 37.54±4.96 | **47.04±2.77** | **45.72±7.22** |
| medium-replay | hopper | 23.23±2.71 | 35.24±36.73 | **79.60±18.46** | 30.72±10.57 |
| medium-replay | walker2d | -0.23±0.02 | 1.58±3.06 | **71.20±37.33** | 6.64±1.82 |
| medium-expert | halfcheetah | 88.46±2.19 | 88.00±6.00 | **96.38±0.82** | 85.87±8.19 |
| medium-expert | hopper | 58.91±16.27 | **91.04±29.10** | 74.97±29.34 | 2.81±1.47 |
| medium-expert | walker2d | **109.05±0.69** | **109.61±0.29** | **110.00±0.36** | 0.64±1.64 |
| expert | halfcheetah | 88.15±4.16 | 93.87±0.62 | **99.16±1.09** | 48.57±14.05 |
| expert | hopper | **95.63±24.05** | **96.38±4.84** | 70.74±18.13 | 2.12±1.84 |
| expert | walker2d | **108.68±0.49** | **109.57±0.32** | **110.52±0.25** | 24.42±23.81 |
| full-replay | halfcheetah | 15.65±12.65 | 73.29±2.56 | **76.60±0.65** | **80.54±0.97** |
| full-replay | hopper | 91.48±32.87 | **106.67±0.65** | **107.07±0.51** | 48.80±20.94 |
| full-replay | walker2d | 4.27±1.86 | 99.16±0.75 | **102.15±1.02** | 38.84±26.75 |
| random | halfcheetah | 20.57±1.93 | 24.27±1.56 | 29.87±2.06 | **32.31±1.52** |
| random | hopper | 7.64±0.19 | 7.62±0.20 | **11.08±2.51** | 9.64±0.72 |
| random | walker2d | -0.09±0.00 | -0.07±0.01 | **9.18±4.38** | 5.34±2.17 |
| Average | | 49.98 | 66.13 | **73.63** | 29.73 |

Table 13: Final performance of TD3-AN with hybrid noise distribution across different $\log \sigma$ values in Gym-MuJoCo environments. Results are averaged over five training seeds with 10 trajectories per seed. Standard deviations are indicated by $\pm$, and the highest scores within 5% of the best per task are highlighted in bold.

| Dataset | Environment | -20 | -10 | -5 | -1 |
|---|---|---|---|---|---|
| medium | halfcheetah | **61.49±0.73** | **60.30±0.84** | 56.00±0.41 | 25.04±7.55 |
| medium | hopper | 91.46±8.23 | **95.66±2.80** | **98.18±2.07** | 65.68±1.60 |
| medium | walker2d | 39.52±31.27 | 76.67±34.27 | **88.49±1.28** | 38.13±35.15 |
| medium-replay | halfcheetah | **53.35±0.68** | **52.25±0.75** | 47.39±2.42 | 13.92±11.19 |
| medium-replay | hopper | 96.04±7.99 | **102.30±0.41** | **100.31±0.93** | 0.95±0.57 |
| medium-replay | walker2d | **84.03±16.25** | **86.30±8.64** | **87.82±14.09** | 0.86±1.61 |
| medium-expert | halfcheetah | **94.05±6.34** | **96.44±1.83** | **91.66±4.29** | 43.59±3.74 |
| medium-expert | hopper | 41.61±8.27 | 71.33±9.41 | 83.46±18.92 | **108.84±2.10** |
| medium-expert | walker2d | **114.85±0.47** | **111.98±0.90** | **111.27±0.49** | 104.10±10.82 |
| expert | halfcheetah | **104.45±8.52** | 97.84±13.39 | **101.62±1.00** | 73.68±8.91 |
| expert | hopper | 37.20±11.38 | 48.20±8.17 | 65.58±9.04 | **109.03±6.74** |
| expert | walker2d | 32.95±55.54 | 59.99±63.97 | **112.76±0.45** | **108.92±0.33** |
| full-replay | halfcheetah | **81.17±1.07** | **81.12±0.63** | **78.61±1.21** | -1.37±0.51 |
| full-replay | hopper | **108.31±0.33** | **107.67±1.13** | **106.86±0.31** | 0.86±0.53 |
| full-replay | walker2d | 102.16±7.32 | **103.78±1.60** | **103.11±0.86** | 0.67±1.21 |
| random | halfcheetah | **30.01±1.12** | **29.49±1.20** | 25.62±0.89 | 2.22±1.17 |
| random | hopper | **9.55±0.99** | **9.99±1.44** | 9.09±0.65 | 1.04±0.06 |
| random | walker2d | **5.69±1.82** | **5.79±1.66** | 3.58±2.69 | -0.14±0.02 |
| Average | | 66.00 | 72.06 | **76.19** | 38.67 |

Table 14: Final performance of IQL-AN with $\tau = 0.7$ in Gym-MuJoCo environments. Results are averaged over five training seeds with 10 trajectories per seed. Standard deviations are indicated by $\pm$, and the highest scores within 5% of the best per task are highlighted in bold.

| Dataset | Environment | -20 | -10 | -5 | -1 |
|---|---|---|---|---|---|
| medium | halfcheetah | **55.41**±0.71 | **54.04**±0.47 | 51.79±0.32 | 44.87±0.24 |
| medium | hopper | 68.16±5.66 | 92.04±10.86 | **98.41**±2.57 | 65.48±4.48 |
| medium | walker2d | **87.47**±8.21 | **86.32**±4.76 | 79.93±17.56 | 78.43±4.73 |
| medium-replay | halfcheetah | **49.45**±0.97 | 45.65±2.59 | 43.14±3.23 | 26.95±6.87 |
| medium-replay | hopper | **100.82**±0.86 | **100.25**±0.43 | **99.63**±0.49 | 3.69±3.34 |
| medium-replay | walker2d | **85.07**±14.89 | 79.22±41.81 | **88.76**±8.04 | 9.49±6.16 |
| medium-expert | halfcheetah | 25.91±3.30 | 29.72±4.85 | 35.72±6.05 | **69.73**±22.27 |
| medium-expert | hopper | 31.52±19.68 | 46.45±21.39 | 76.87±29.94 | **105.26**±8.35 |
| medium-expert | walker2d | 101.61±16.03 | **109.62**±1.29 | 106.17±7.39 | **108.31**±0.75 |
| expert | halfcheetah | 9.73±3.00 | 9.90±5.86 | 10.71±2.16 | **75.79**±24.55 |
| expert | hopper | 11.51±12.58 | 37.40±26.85 | 69.05±10.55 | **108.61**±5.87 |
| expert | walker2d | 51.41±27.97 | 40.72±35.29 | 98.75±19.89 | **108.17**±0.52 |
| full-replay | halfcheetah | **78.28**±0.28 | **76.58**±0.87 | 75.62±1.44 | 65.04±2.21 |
| full-replay | hopper | **105.61**±0.67 | **105.78**±0.80 | **105.86**±0.80 | 104.42±0.92 |
| full-replay | walker2d | 97.35±6.67 | **100.56**±3.81 | 99.17±2.62 | 69.59±13.45 |
| random | halfcheetah | **24.25**±7.02 | **23.74**±7.43 | 16.76±4.99 | 1.71±0.92 |
| random | hopper | **8.78**±0.43 | **8.68**±1.24 | **9.04**±0.50 | 0.86±0.05 |
| random | walker2d | **19.14**±7.39 | **19.41**±6.36 | -0.11±0.00 | -0.21±0.00 |
| Average | | 56.19 | 59.23 | **64.74** | 58.12 |

Table 15: Final performance of TD3-AN in AntMaze environments. Results are averaged over five training seeds with 100 trajectories per seed. Standard deviations are indicated by $\pm$, and the highest scores within 5% of the best per task are highlighted in bold.

| Dataset | Environment | $\alpha = 0.3$ | | | $\alpha = 0.5$ | | | $\alpha = 1.0$ | | |
|---|---|---|---|---|---|---|---|---|---|---|
| | | -20.0 | -10.0 | -5.0 | -20.0 | -10.0 | -5.0 | -20.0 | -10.0 | -5.0 |
| - | umaze | 90.80±3.27 | **96.80**±1.30 | **93.80**±2.77 | **93.60**±4.88 | **97.20**±2.95 | **98.40**±1.52 | **98.40**±1.14 | 97.80±3.27 | **98.40**±1.52 |
| diverse | umaze | 55.40±33.88 | 31.20±26.44 | 25.60±13.35 | 64.20±15.96 | 40.60±20.23 | 27.80±6.10 | **74.60**±10.95 | 50.00±23.61 | 39.20±3.49 |
| play | medium | 78.20±6.34 | 77.20±4.32 | 55.20±26.52 | 65.80±9.42 | 76.00±6.20 | 75.40±5.86 | 68.80±13.55 | **83.80**±6.06 | 73.40±10.95 |
| diverse | medium | 74.40±11.55 | **82.00**±9.62 | 20.40±17.49 | 61.00±25.50 | 79.20±6.83 | 11.80±13.66 | **82.60**±6.88 | **85.80**±2.59 | 12.40±7.57 |
| play | large | 62.00±17.13 | 44.60±16.29 | 17.00±11.29 | **65.40**±6.15 | 52.20±10.78 | 19.80±12.03 | 57.80±9.98 | 43.40±12.58 | 14.00±9.27 |
| diverse | large | 39.80±11.45 | 45.40±11.89 | 27.80±11.82 | 42.20±21.50 | **56.20**±4.21 | 35.80±19.23 | 53.00±21.35 | **58.00**±11.38 | 23.40±17.83 |
| Average | | 66.77 | 62.87 | 39.97 | 65.37 | 66.90 | 44.83 | 72.53 | 69.80 | 43.47 |

Table 16: Final performance of IQL-AN in AntMaze environments. Results are averaged over five training seeds with 100 trajectories per seed. Standard deviations are indicated by $\pm$, and the highest scores within 5% of the best per task are highlighted in bold.

| Dataset | Environment | $\alpha = 0.3$ | | $\alpha = 0.5$ | | $\alpha = 1.0$ | |
|---|---|---|---|---|---|---|---|
| | | -20.0 | -10.0 | -20.0 | -10.0 | -20.0 | -10.0 |
| - | umaze | **89.40**±1.52 | **89.20**±1.79 | **90.00**±3.32 | 88.20±2.49 | **89.80**±1.30 | **91.20**±2.39 |
| diverse | umaze | 61.60±2.97 | 53.60±6.07 | 62.60±2.79 | 59.80±2.86 | **68.00**±7.18 | 60.80±6.18 |
| play | medium | **74.40**±8.23 | **73.00**±3.46 | **74.20**±4.92 | 70.40±4.62 | 55.40±5.86 | 50.00±4.69 |
| diverse | medium | 68.50±5.97 | **75.20**±3.63 | 70.80±4.15 | 67.20±5.67 | 49.40±3.36 | 42.40±9.71 |
| play | large | **49.40**±6.73 | **48.60**±4.45 | 38.80±2.77 | 35.20±4.44 | 2.60±3.13 | 3.00±1.22 |
| diverse | large | **52.80**±5.89 | **52.20**±6.38 | 43.00±10.61 | 36.40±5.08 | 3.40±2.51 | 1.60±1.67 |
| Average | | 66.02 | 65.30 | 63.23 | 59.53 | 44.77 | 41.50 |

## C.6 TRAINING CURVES

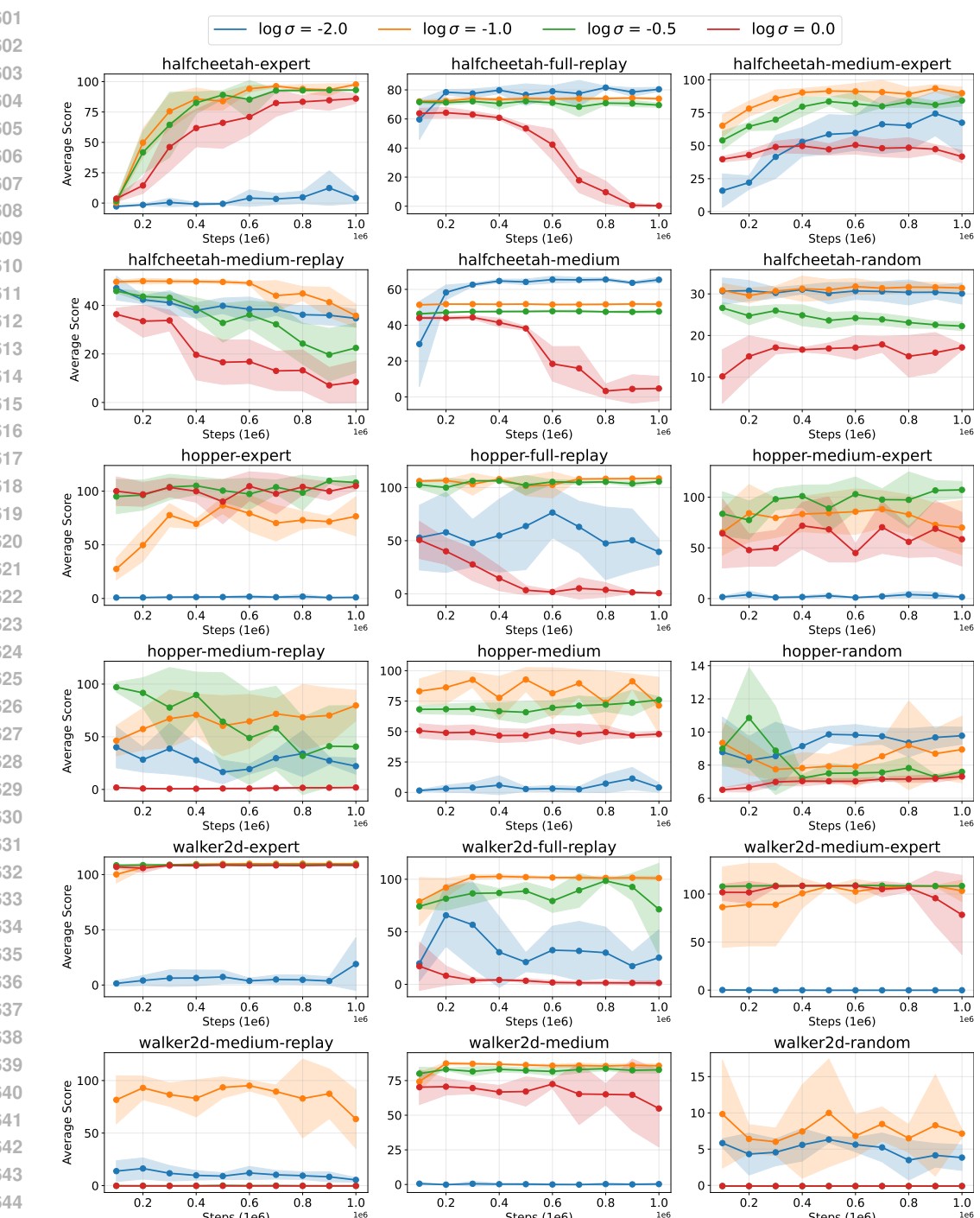

Figure 7: Training curves of TD3-AN with Gaussian Noise Distribution in Gym-MuJoCo.

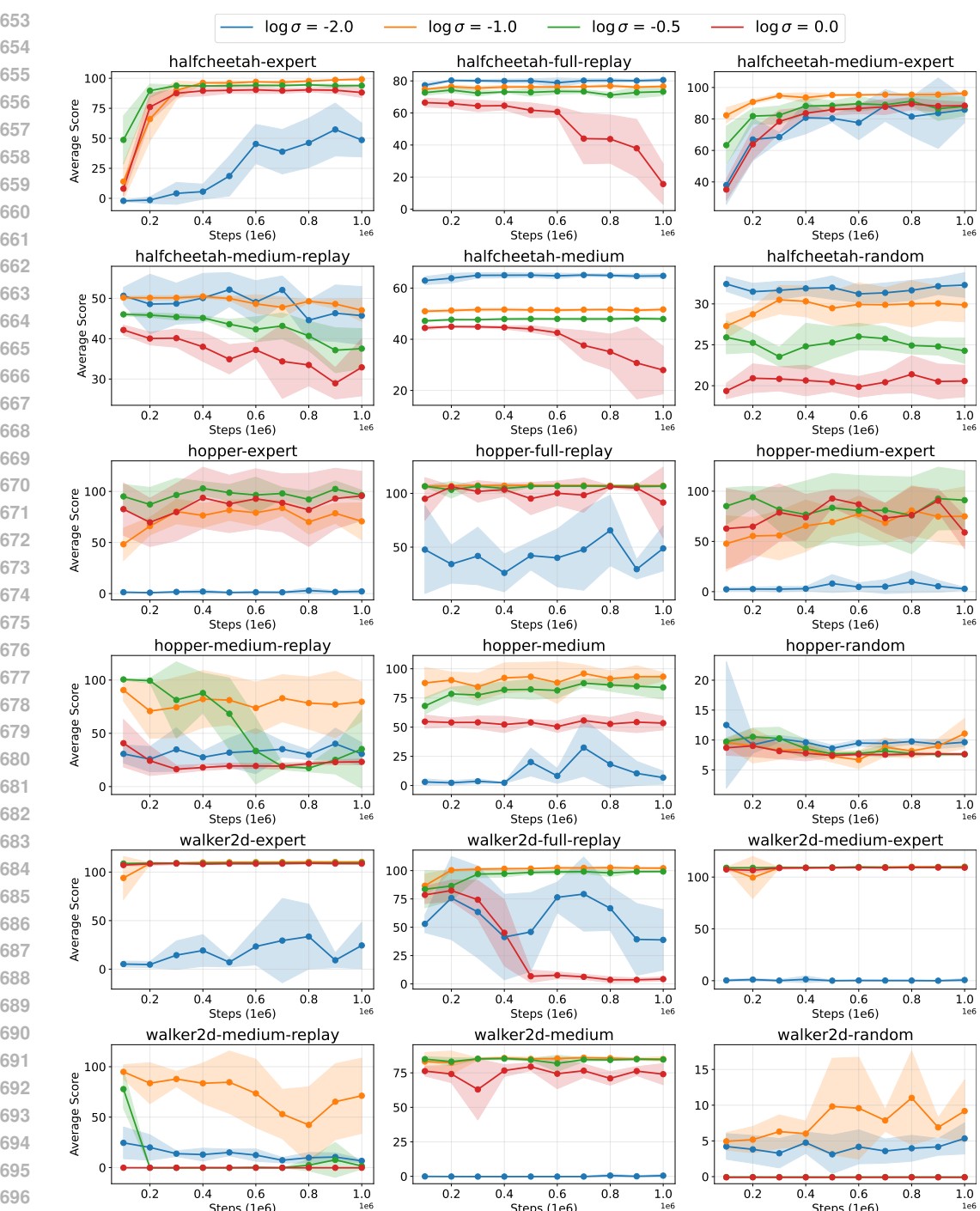

Figure 8: Training curves of TD3-AN with Laplace Noise Distribution in Gym-MuJoCo.

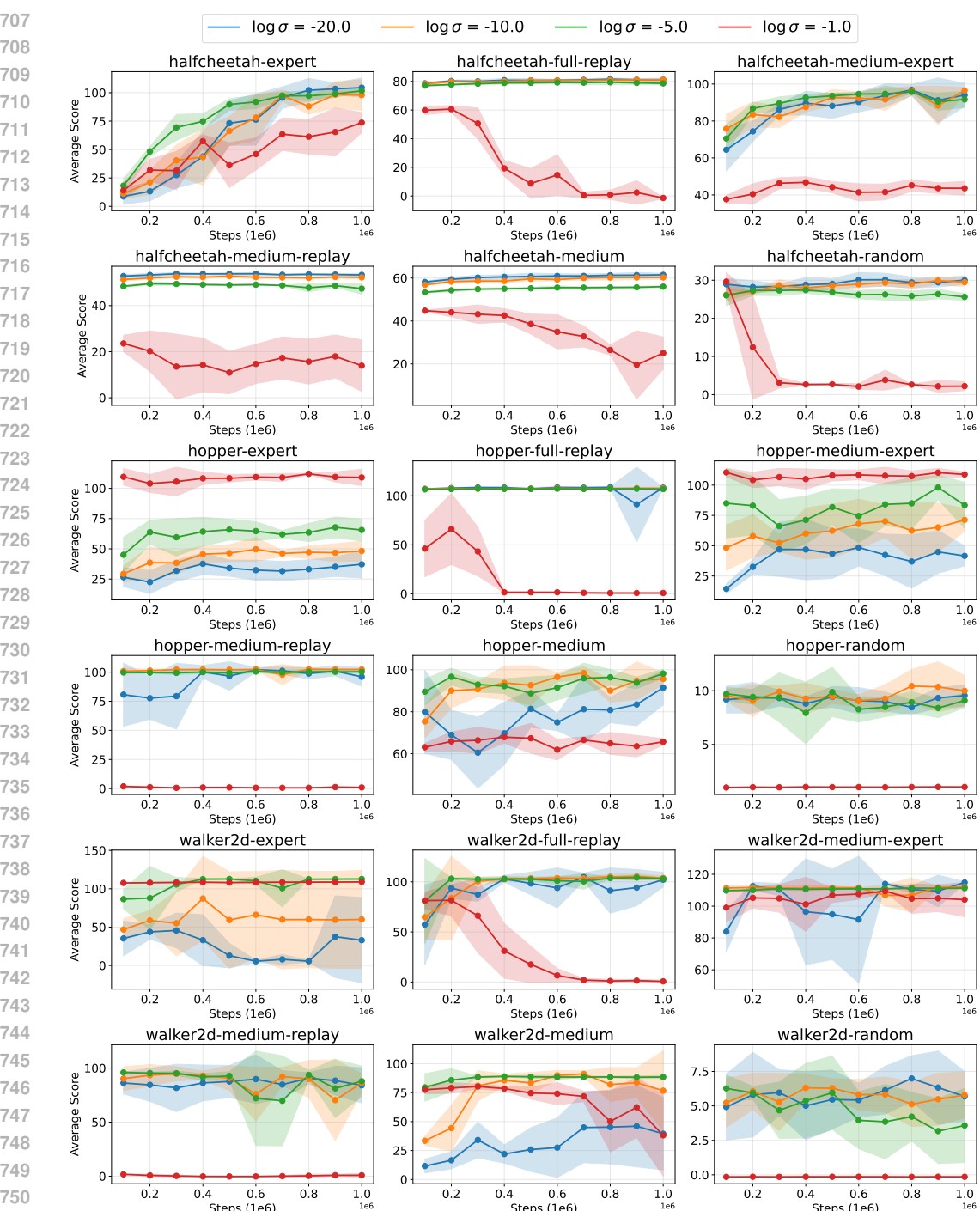

Figure 9: Training curves of TD3-AN with hybrid noise distribution in Gym-MuJoCo.

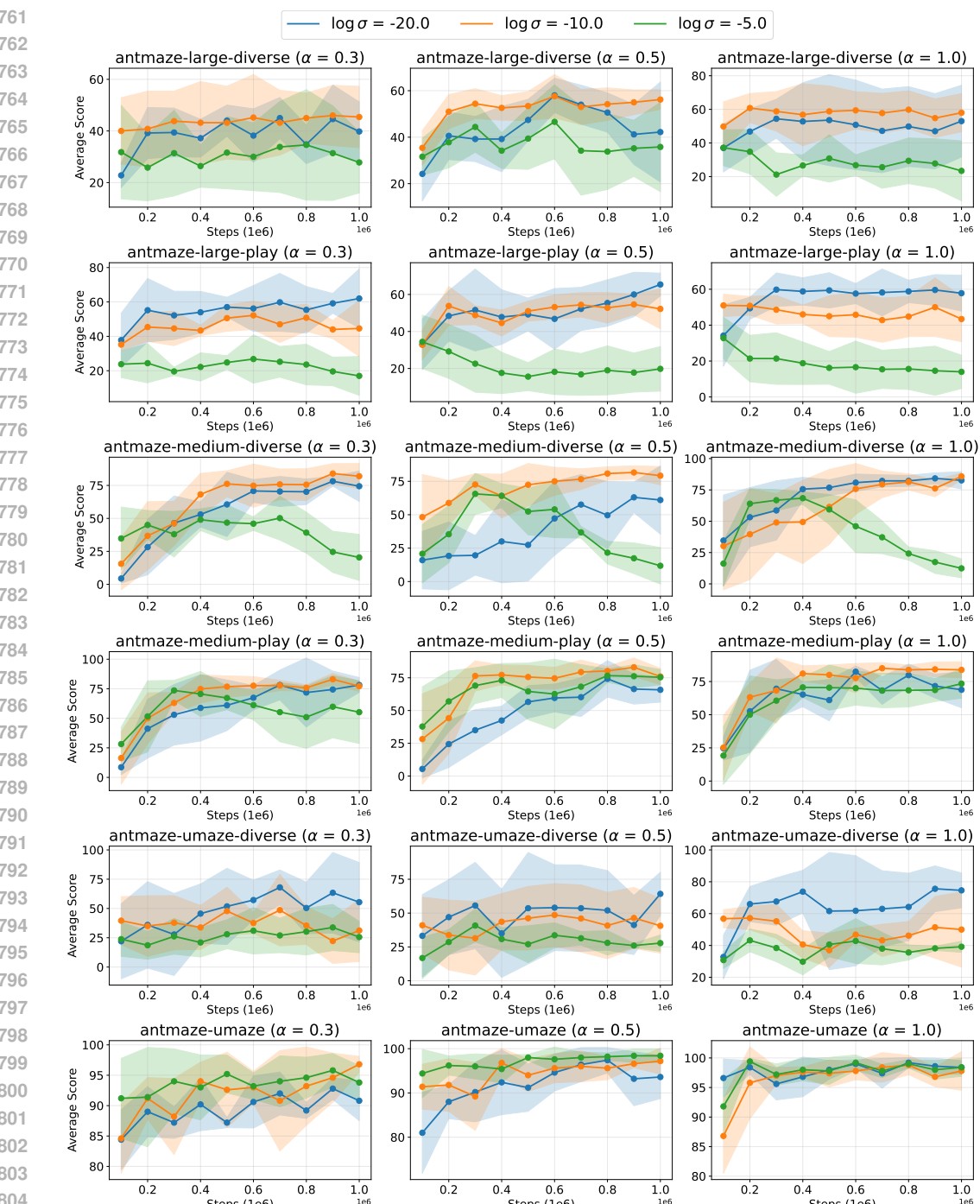

Figure 10: Training curves of TD3-AN with hybrid noise distribution in AntMaze.

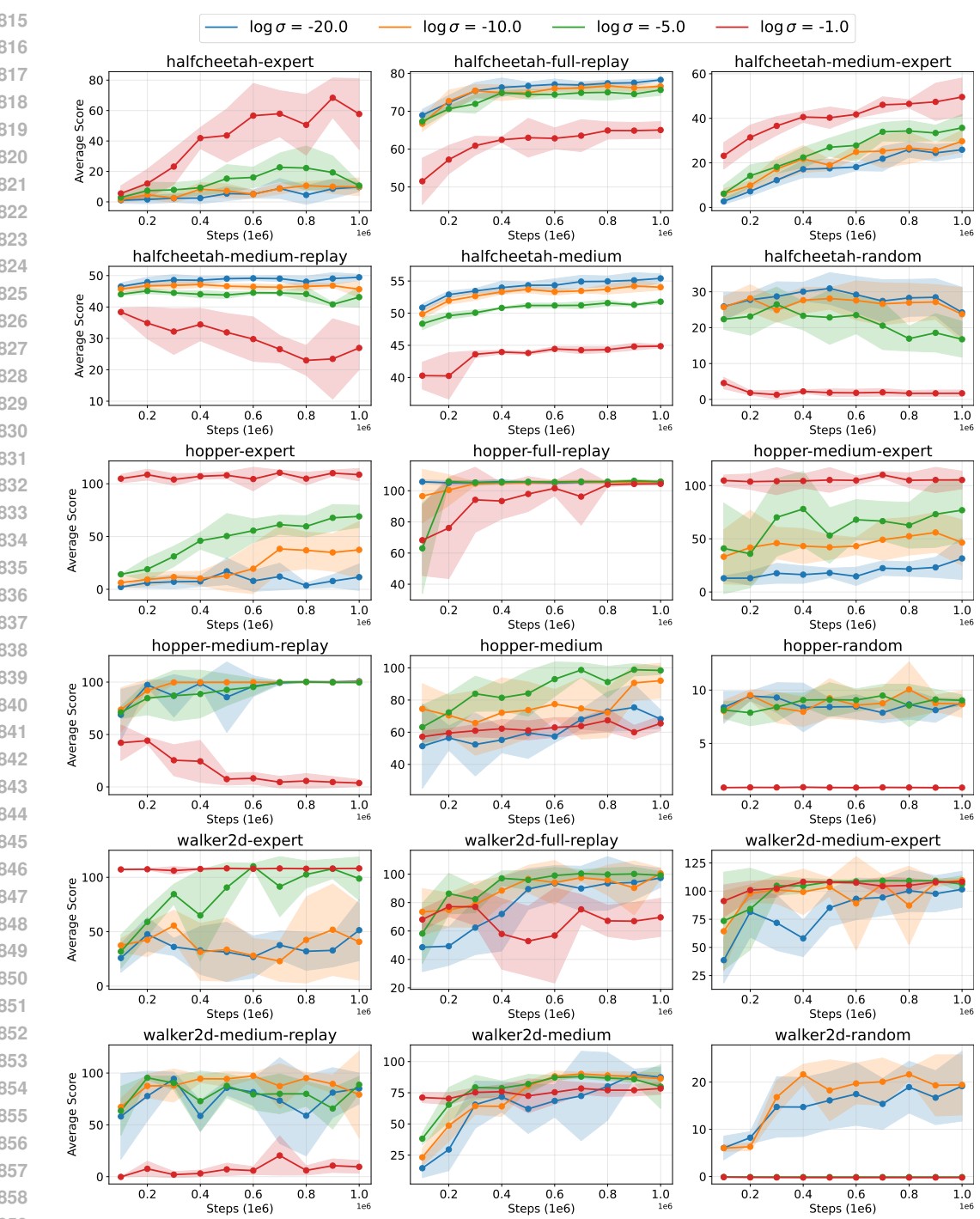

Figure 11: Training curves of IQL-AN with hybrid noise distribution in Gym-MuJoCo ($\tau = 0.7$).

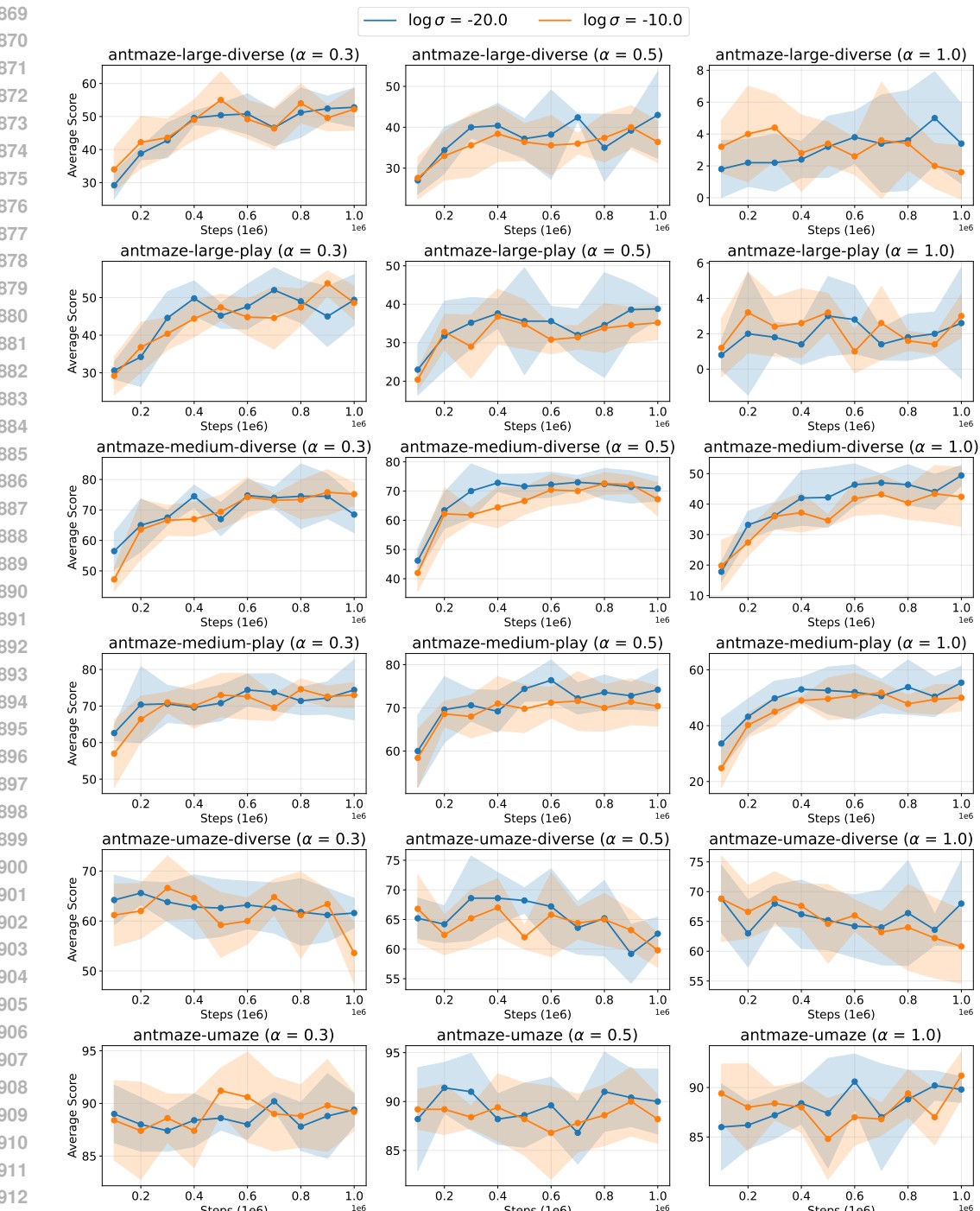

Figure 12: Training curves of IQL-AN with hybrid noise distribution in AntMaze.

