# OpenReview forum: "Offline Reinforcement Learning with Penalized Action Noise Injection"
_ICLR.cc/2026/Conference — ICLR 2026 Conference Withdrawn Submission_

### Official Review · Reviewer_SH7n · 2025-10-23

**Soundness:** 2
**Presentation:** 3
**Contribution:** 2
**Rating:** 2
**Confidence:** 5

**Summary:**

This paper introduces Penalized Action Noise Injection (PANI), a simple and broadly applicable method for offline reinforcement learning that mitigates overestimation of out-of-distribution actions by injecting controlled noise into dataset actions and penalizing Q-values proportionally to the noise magnitude. Theoretically grounded as Q-learning on a Noisy Action MDP, PANI requires minimal modification to existing algorithms and delivers substantial performance gains across benchmarks as an effective drop-in enhancement.

**Strengths:**

## Strengths

- This paper is easy to read and easy to follow
- This paper conducts experiments on numerous datasets and tasks, including D4RL locomotion datasets, Adroit datasets, AntMaze datasets, and OGBench datasets
- This paper includes detailed learning curves in the Appendix, which can be helpful for readers to understand the hyperparameter sensitivity of the proposed method

**Weaknesses:**

## Weaknesses

- No codes are attached. It is not clear whether the results reported in the main text and the appendix are reproducible. The authors include the anonymous code link in Appendix C, but it does not work. Hence, it is difficult to judge the effectiveness of the AN method
- There are numerous incomplete sentences in this paper, e.g.,
  - Line 40, *Although these methods achieve strong empirical results.*
  - Line 322, *Q2. Is PANI computationally more efficient than diffusion-based methods?* The authors do not seems to compare the training time of PANI against diffusion-based methods and I do not see any reason for such a comparison, considering the fact that PANI itself is not a diffusion-based method
- No related work discussion is included in the main text. The authors defer the related work part to the appendix. The related work part should appear in the main text. Furthermore, the related work discussion is insufficient. The authors should cite more previous offline RL and recent offline RL papers, both model-free ones and model-based ones
- I actually cannot understand the necessity of Section 3 and Figure 1. The overestimation issue (or the extrapolation error) is a well-known problem in offline RL (see BCQ [1] and BEAR [2]). I do not think there is any need for introducing another toy example to show this. Also, the authors do not provide sufficient information about the Ring and Pinwheel datasets. Since there are no new results in the overestimation issue, I do not think Section 3 and Figure 1 are necessary. There is also no need to introduce the DSM objective. If the authors would like to introduce some background knowledge about the diffusion model, they should only briefly introduce Lines 101-123 in the main text, though this paper does not seem to be related to diffusion models.
- The proposed method shares many similarities with some prior works, but the authors do not compare against those methods or discuss this in the main text or the appendix
  - The proposed method injects action noise into the learning objectives to avoid overestimation, which can be similar to robust offline RL. The authors cite RORL [3] but do not discuss it. Intuitively, enhancing the robustness of the offline RL agent should achieve a similar effect as PANI. The performance of PANI seems to be inferior to RORL.
  - The proposed method ultimately subtracts the action deviation terms in the target value, which can be similar to anti-exploration methods like SAC-RND [4], SAC-DRND [5]. The authors should include a comprehensive discussion of those methods and empirically compare PANI against them
  - According to Lines 148-149, the proposed method trains with $Q(s,\bar{a})$ rather than $Q(s,a)$, i.e., it actively trains samples that may lie outside of the static offline dataset. This reminds me of a series of works that construct pseudo targets for OOD actions, including EPQ [6], MCQ [7], and PBRL [8]. These should be discussed and empirically compared
- Furthermore, I unfortunately have some concerns about the proposed method itself, i.e.,
  - In Lines 148-149, the target value is penalized with $\\|a - \bar{a}\\|^2_2$. This can be problematic since different tasks can have different reward scales and action scales. It is highly possible that the penalty term $\\|a - \bar{a}\\|^2_2$ does not have any effect on the final reward. I am a bit confused that it turns out that PANI incurs good performance. Can the authors elaborate more on explaining the reasons behind this? Also, I do not think that PANI is general enough to be applied to other datasets when the reward scale is large, while the actions are normalized to lie in $[-1,1]$. Any comments here? I am also curious about why the authors do not adopt $\alpha\_{\rm inject}\\|a - \bar{a}\\|^2_2$ as the penalty term, where $\alpha\_{\rm inject}$ is the hyperparameter that controls the scale of the penalty.
  - In Lines 292-293, the authors adopt the hybrid noise distribution trick where the distribution is a convex combination of the uniform distribution and $q_t$. It is unclear why uniform distribution is adopted here. The theoretical analysis does not explain the validity of the hybrid noise distribution. Why is uniform distribution necessary here, and why can uniform distribution help? I would expect some theoretical insights or stronger evidence in explaining this.
  - In Figure 3, it is super clear that Laplace distribution is better than Gaussian distribution, but the authors wrote in Line 308 that *In our experiments, we used a Gaussian-base noise*. I think this is contradictory to previous experiments and analysis.
- Some of the compared baselines are weak. Meanwhile, the good performance of PANI seems to be the result of careful hyperparameter tuning based on the appendix. Notably, the hyperparameter range can be quite large (as shown in Table 4-8). It raises concerns about the advantages of the proposed method on other datasets. The authors adopt 20000 steps for Adroit tasks, which can be very problematic. Based on my own experience, the performance of offline RL agents on pen tasks can be quite good when the training steps are few, but the performance can drop drastically when the steps are large (e.g., 1e6). Furthermore, the authors also carefully tune the hyperparameters of the base offline RL algorithm on each task, which can incur unfair performance comparison

There are also some minor points,
- The authors claim that they use TD3 as the base algorithm when combining with AN, but it turns out that they still use TD3+BC (c.f. Equation 70 in Appendix C.2). This can be confusing
- Numerous equations in the main text are not numbered
- Some of the experiments are conducted under only 3 random seeds

Overall, I believe that this manuscript is not fully ready for publication and needs to be significantly revised.

## References

[1] Off-Policy Deep Reinforcement Learning without Exploration. ICML

[2] Stabilizing Off-Policy Q-Learning via Bootstrapping Error Reduction. NeurIPS

[3] RORL: Robust Offline Reinforcement Learning via Conservative Smoothing. NeurIPS

[4] Anti-exploration by random network distillation. ICML

[5] Exploration and anti-exploration with distributional random network distillation. ICML

[6] Exclusively penalized q-learning for offline reinforcement learning. NeurIPS

[7] Mildly conservative q-learning for offline reinforcement learning. NeurIPS

[8] Pessimistic bootstrapping for uncertainty-driven offline reinforcement learning. ICLR

**Questions:**

- Figure 3 is a bit confusing. How do you construct the distributions? What does target mean? It is quite hard to interpret the figure
- I doubt the reported training wall-clock time in Figure 4(b). It is impossible for IQL to achieve a runtime of 9m 33s for 1e6 steps even with Jax. In the original IQL paper, the runtime with Jax is 20min. How could the training time for IQL and TD3+BC be so short?

---

### Official Review · Reviewer_mqZV · 2025-10-26

**Soundness:** 2
**Presentation:** 3
**Contribution:** 2
**Rating:** 4
**Confidence:** 3

**Summary:**

PANI mitigates OOD Q-value overestimation in offline RL by injecting noise into actions and penalizing deviations in the Q-learning target. This broadens action space exploration and reduces reliance on neural network generalization. Theoretically, PANI is equivalent to Q-learning in a Noisy Action MDP. It's a lightweight, algorithm-agnostic method that significantly boosts performance for algorithms like IQL and TD3 on D4RL benchmarks.

**Strengths:**

1. The paper is easy to follow.
2. The theoretical proof is sound, and experiments are strong in some way.
3. The algorithm is simple. It is a lightweight, "drop-in" modification requiring only minimal changes to the standard Q-update step of existing algorithms.

**Weaknesses:**

1. The practical implementation for real-world cases is limited. As we can imagine, in some critical scenarios, a small perturbation to actions will cause catastrophic failure.
2. In Table 1, PANI shows significant gains on older diffusion-free methods like TD3/IQL, but with limited comparison to QGPO, especially in challenges AntMaze tasks in Table 1. Could the author attempt to apply PANI to more advanced algorithms to demonstrate improvements over QGPO?
3. For Table 3, I am confused why the baselines are totally different from Table 1. Could the authors explain the reason and maintain the same baselines in Table 1?

Minors:
1. In Section 4, what is $y$, this should be clearly written. Is this related to Q or V-function?

**Questions:**

1. Regarding the application of PANI to IQL: IQL is generally understood as an in-sample learning method without out-of-distribution (OOD) actions. Given that PANI is also designed to address OOD overestimation, could the authors please elaborate on why it provides such a significant performance improvement to IQL? Is the primary benefit from regularizing the Q-function landscape in the local neighborhood of dataset actions, even if these specific OOD points are not explicitly queried by the in-sample learning paradigm? (This is my most important question.)
2. The penalty term $\|a-\bar{a}\|^2$ is central to the NAMDP formulation. What is the intuition for choosing this specific squared L2 norm? Have you experimented with alternative penalty functions, such as an L1 norm or a penalty that is weighted by the Q-value or the data distribution density?

---

### Official Review · Reviewer_nUxv · 2025-10-31

**Soundness:** 2
**Presentation:** 3
**Contribution:** 2
**Rating:** 2
**Confidence:** 4

**Summary:**

The paper proposes Penalized Action Noise Injection (PANI), a general method to improve offline reinforcement learning (RL) by learning pessimistic value estimations on out-of-distribution (OOD) actions. PANI injects noise into the dataset actions and penalizes deviations from them. The reward penalty increases while the sampled action is further from the dataset action, therefore, to mitigate overestimation for OOD actions. Experiments on D4RL benchmarks demonstrate that PANI improves the performance of existing offline RL algorithms.

**Strengths:**

The paper clearly explains the intuition and methodology of the proposed algorithm. Empirical evaluations on effectiveness and ablation studies are provided. The introduction of the Noisy Action MDP provides a principled explanation for why penalized noisy updates improve robustness to OOD actions.

The method is straightforward, general, and easy to integrate into existing offline RL algorithms, requiring minimal modification.

Adjusting the penalty according to the distance between the noisy and dataset actions intuitively promotes smoother Q-value surfaces, helping to reduce sharp value changes around the data boundary.

Ablation studies and distributional analyses are presented to support the design choices, including the selection of different noise distributions (Gaussian, Laplace, and Hybrid).

**Weaknesses:**

When the dataset covers a very narrow action distribution while the action space is large, the injected noise may still fail to expose the Q-function to sufficiently diverse actions, or take extra learning time to sample a sufficient number of actions to represent the OOD action space. While the authors partly address this by using hybrid noise distributions, a more detailed discussion on how PANI behaves with highly concentrated expert datasets would strengthen the paper.

The paper could benefit from stronger baselines for comparison. In particular, multiple other works in literature also introduce pessimistic or OOD-aware value regularization [1][2][3][4]. Works following a similar idea should be included as baselines to better position PANI among similar approaches in terms of performance, computational cost, and wall-clock time.

As shown in prior works [2][3], pessimistic regularization can sometimes lead to overly low value estimates. PANI does not explicitly constrain the lower bound of Q-values. The paper would benefit from discussing whether this issue arises in the experiments and how it might be mitigated.

[1] Kumar, Aviral, et al. "Conservative q-learning for offline reinforcement learning." Advances in neural information processing systems 33 (2020)

[2] Nakamoto, Mitsuhiko, et al. "Cal-ql: Calibrated offline rl pre-training for efficient online fine-tuning." Advances in Neural Information Processing Systems 36 (2023)

[3] Lyu, Jiafei, et al. "Mildly conservative q-learning for offline reinforcement learning." Advances in Neural Information Processing Systems 35 (2022)

[4] Kim, Jeonghye, et al. "Penalizing Infeasible Actions and Reward Scaling in Reinforcement Learning with Offline Data." Forty-second International Conference on Machine Learning (2025).

**Questions:**

Since higher noise levels make the NAMDP diverge from the original MDP (lines 230-231), does the resulting optimal policy risk lie outside the dataset coverage? If so, would it be possible that this amplifies OOD sampling issues during Q function bootstrapping, as the Q function will still update towards an inaccurate estimation?

---

### Official Review · Reviewer_m2Fr · 2025-11-01

**Soundness:** 3
**Presentation:** 3
**Contribution:** 3
**Rating:** 6
**Confidence:** 4

**Summary:**

This manuscript proposes the PANI method to solve the OOD issue in offline RL. The PANI method is a simple and efficient Q-learning method by injecting controlled noise to mitigate overestimation. The authors also propose Noise Action MDP and construct the theoretical foundation for the method. The PANI is applied with various offline RL benchmarks, and experiments demonstrate the superiority of the proposed method.

**Strengths:**

* The OOD issue, are classic topics in offline RL, it is appreciated that the authors consider this issue from the new perspectives. The PANI is compatible with other methods and this merit is quite appealing.

* It is also appreciated that the authors formalize the framework of PANI with Noise Action MDP, as well as proposing the concept of hybrid noise distribution,  which builds the theoretical foundation for the entire methodology.

* The reflected flow noise generator can produce complex multimodal noise, which is helpful for some scenarios where real actions distribution are quite complex.

**Weaknesses:**

1. Some related references are missing, and it is suggested to consider the related work in the manuscript.

* https://arxiv.org/abs/2202.06239

* https://arxiv.org/abs/1911.11361

* https://ieeexplore.ieee.org/document/10432784

* https://arxiv.org/abs/2301.12130

2. From the experiments, it seems the TD3-AN always performs better than IQL-AN, is there any furhter explanations on this phenomenon? For Antmaze benchmark, it seems IQL-AN is inferior to most alternatives, it is suggested that the authors provide more analysis on the experiment results.

3. For the noise distribution, which is a critical topic to discuss. The authors provide the ablation studies in experiments, however, is there any theoretical analysis showing what kind of distribution property is the key to a robust performance? Such as the skewness or kurtosis. In addition, is the proposed hybrid noise distribution suitable for discrete action scenarios?

4. Although it is mentioned that PANI has low computational cost, it does not analyze the computational efficiency in high-dimensional action spaces. For example, when the action dimension increases 20 (HalfCheetah), the sampling and gradient calculation time of the hybrid noise distribution will increase significantly, how does the training cost/time of PANI increase?

**Questions:**

See the weakness above

---

### Note · Authors · 2025-11-19

**Comment:**

We sincerely appreciate the reviewers' thorough and insightful feedback. We fully acknowledge that substantial revisions are necessary, and we are committed to addressing all comments with great care. By incorporating the suggestions provided, we aim to significantly improve the quality of the manuscript and resubmit a much-strengthened version in the future.

**Withdrawal Confirmation:**

I have read and agree with the venue's withdrawal policy on behalf of myself and my co-authors.